# XR-1: Towards Versatile Vision-Language-Action Models via Learning Unified Vision-Motion Representations

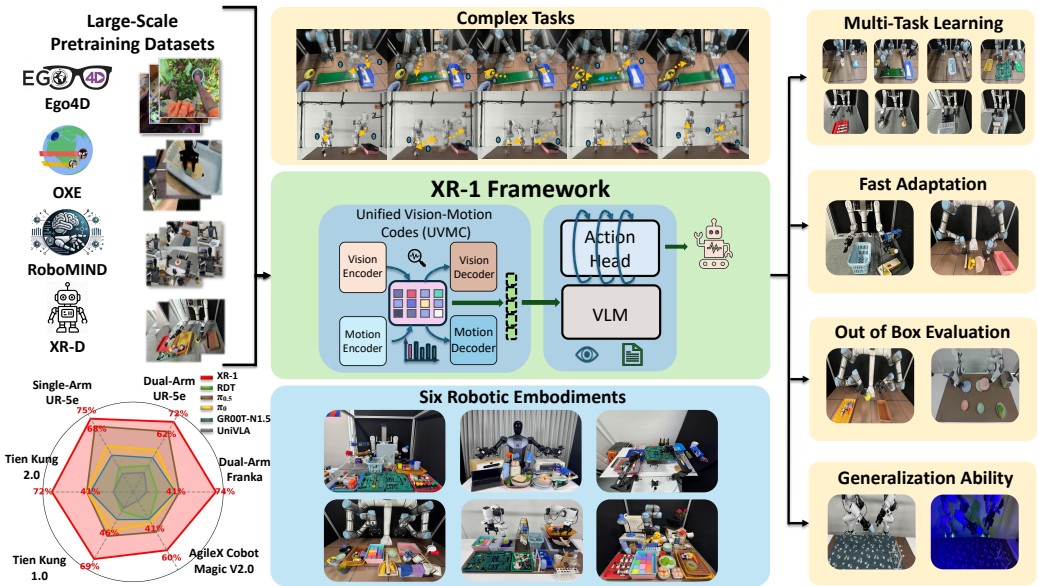

Figure 1: We introduce **X Robotic Model 1 (XR-1)**, a versatile and scalable vision-language-action framework. XR-1 supports robust multi-task learning across diverse robot embodiments and environments.

## Abstract

Recent progress in large-scale robotic datasets and vision-language models (VLMs) has advanced research on vision-language-action (VLA) models. However, existing VLA models still face two fundamental challenges: (*i*) producing precise low-level actions from high-dimensional observations, (*ii*) bridging domain gaps across heterogeneous data sources, including diverse robot embodiments and human demonstrations. Existing methods often encode latent variables from either visual dynamics or robotic actions to guide policy learning, but they fail to fully exploit the complementary multi-modal knowledge present in large-scale, heterogeneous datasets. In this work, we present **X Robotic Model 1 (XR-1)**, a novel framework for versatile and scalable VLA learning across diverse robots, tasks, and environments. At its core, XR-1 introduces the *Unified Vision-Motion Codes (UVMC)*, a discrete latent representation learned via a dual-branch VQ-VAE that jointly encodes visual dynamics and robotic motion. UVMC addresses these challenges by (*i*) serving as an intermediate representation between the observations and actions, and (*ii*) aligning multimodal dynamic information from heterogeneous data sources to capture complementary knowledge. To effectively exploit UVMC, we propose a *three-stage training paradigm*: (*i*) self-supervised UVMC learning, (*ii*) UVMC-guided pretraining on large-scale cross-embodiment robotic datasets, and (*iii*) task-specific post-training. We validate XR-1 through extensive real-world experiments with more than 14,000 rollouts on six different robot embodiments, spanning over 120 diverse manipulation

tasks. XR-1 consistently outperforms state-of-the-art baselines such as $\pi_{0.5}$, $\pi_0$, RDT, UniVLA, and GR00T-N1.5 while demonstrating strong generalization to novel objects, background variations, distractors, and illumination changes. Our project is at https://xr-1-vla.github.io/.

# 1 INTRODUCTION

The long-term goal of Embodied Artificial Intelligence (Embodied AI) (Pfeifer & Iida, 2004) is to build general-purpose robotic agents capable of following natural language instructions to perform diverse tasks in real-world environments, ranging from households and factories to hospitals and laboratories. Recent progress in Vision-Language Models (VLMs) (Bai et al., 2023; Gao et al., 2024; Li et al., 2023; Liu et al., 2023; Zhang et al., 2024; Beyer et al., 2024; Wang et al., 2024b) has shown that large-scale pretraining on Internet-scale image-text corpora yields strong visual and semantic understanding capabilities. Extending this line, Vision-Language-Action (VLA) models (Zitkovich et al., 2023; Kim et al., 2024; Black et al., 2024; Liu et al., 2025b; Wen et al., 2025a; Cheang et al., 2025b; Liu et al., 2025a; Lee et al., 2025; Intelligence et al., 2025; Bu et al., 2025b) extend VLMs with an action head that grounds perception and language into executable motor commands.

A common training paradigm of VLAs follows a two-stage pipeline: (*i*) large-scale pretraining on cross-embodiment datasets (Walke et al., 2023; O'Neill et al., 2024; Wu et al., 2025a), learning general visuomotor and linguistic priors; and (*ii*) task-specific post-training for a target robot. Despite advances in data-driven learning and the fruitful capacities of VLMs, current VLA models face two key challenges. (*i*) Generating precise low-level actions from high-dimensional observations remains difficult due to the vast search space and inherent multimodal uncertainty. Especially in dexterous or contact-rich tasks, even centimeter-level errors can cause failure. (*ii*) Cross-embodiment datasets utilization is hindered by morphological heterogeneity: robots differ in hardware configuration and degrees of freedom (DoF), while human demonstration videos lack explicit action labels and exhibit appearance discrepancies.

To address these challenges, prior works (Cui et al., 2023; Shafiullah et al., 2022; Lee et al., 2024; Xie et al., 2025; Zheng et al., 2025a) have explored latent representations as intermediate abstractions between observations and actions. One direction encodes robotic action sequences for compact motion modeling (Shafiullah et al., 2022; Wu et al., 2025b; Bauer et al., 2025), but typically requires large labeled datasets that are costly to collect. Another line encodes only visual dynamics from videos (Cui et al., 2023; Hu et al., 2024; Bu et al., 2025a), exploiting abundant video data that contain human demonstrations, but lack explicit action grounding. Both approaches treat vision and action largely in isolation. This separation overlooks the necessity of multimodal alignment: Without integrating visual dynamics and motor actions into a unified space, it is difficult for VLA models to capture coherent task-relevant correspondences across modalities. In contrast, humans naturally fuse heterogeneous sensory inputs into *supramodal codes* (Park et al., 2025), abstracting away embodiment-specific details while preserving task semantics. Inspired by this observation, we argue that effective representation learning for robotics should move beyond unimodal abstractions toward multimodal alignment that jointly encodes visual dynamics and motor control.

Motivated by insights derived from human supramodal cognition and the limitations of prior unimodal representation learning, we propose **X Robotic Model 1 (XR-1)**, a novel framework explicitly designed to achieve cross-data exploitation, cross-modality alignment, and cross-embodiment control. At its core lies the *Unified Vision-Motion Codes (UVMC)*, a discrete latent representation jointly capturing vision dynamics and robotic motion. UVMC is learned via a dual-branch Vector Quantized Variational Autoencoder (VQ-VAE): one branch encodes visual dynamics from raw observations while the other encodes robotic motion. Both share a common codebook in the discrete latent space to enforce unified codes across modalities. To further suppress task-irrelevant visual information and ensure that the vision branch extracts motion-relevant features, we introduce a vision-motion alignment loss that encourages visual codes to be close to their corresponding motion codes. Building upon UVMC, XR-1 employs a three-stage training paradigm: (*i*) self-supervised learning of UVMC on large-scale robotic manipulation datasets together with Internet-scale human demonstration videos; (*ii*) cross-embodiment UVMC-guided pretraining where encoded visuomotor knowledge is injected into the VLM backbone via learnable input tokens; and (*iii*) task-specific

post-training for sharpening performance on particular robots and tasks. This design enables XR-1 to leverage heterogeneous data sources while maintaining embodiment-agnostic consistency.

We extensively evaluate XR-1 through more than 14k rollouts across six distinct robot embodiments, including Tien Kung 1.0/2.0, Single-/Dual-Arm UR-5e, Dual-Arm Franka, and AgileX Cobot Magic 2.0, and covering over 120 manipulation tasks. XR-1 outperforms state-of-the-art baselines such as $\pi_{0.5}$, $\pi_0$, RDT, UniVLA, and GR00T-N1.5 across challenging scenarios involving bimanual collaboration, dexterous manipulation, deformable objects, contact-rich interactions, dynamic settings, and long-horizon manipulation. Our main contributions are summarized as follows:

- We propose **X Robotic Model 1 (XR-1)**, a scalable three-stage training framework for VLA learning that effectively leverages heterogeneous data sources, including Internet-scale human videos and diverse robot datasets, and integrates seamlessly with diverse VLA architectures.
- We introduce the *Unified Vision-Motion Codes (UVMC)*, a discrete latent representation that encodes both environmental dynamics and robotic motion, while an alignment loss enforces consistent multimodal embeddings across embodiments via UVMC.
- We validate XR-1 with over $14,000$ real-world rollouts on six robot embodiments across $123$ tasks, and demonstrate that it consistently outperforms strong baselines such as $\pi_{0.5}$, $\pi_0$, RDT, UniVLA, and GR00T-N1.5.

## 2 RELATED WORK

### 2.1 VISION-LANGUAGE-ACTION MODELS

Developing robust, general-purpose Vision-Language-Action (VLA) policies capable of zero-shot cross-embodiment transfer is a central objective in modern robotics. Initial efforts primarily focused on Imitation Learning (IL) using narrow expert demonstrations (Cui et al., 2023; Zhao et al., 2023; Chi et al., 2023; Ze et al., 2024; Fu et al., 2024; Bharadhwaj et al., 2024; Ze et al., 2024; Cao et al., 2025; Su et al., 2025), which inherently led to limited task scalability and poor generalization across diverse hardware. This bottleneck has been fundamentally addressed by the emergence of large-scale robotic datasets, such as BridgeData (Ebert et al., 2022; Walke et al., 2023), DROID (Khazatsky et al., 2024), Open X-Embodiment (O'Neill et al., 2024), RoboMIND (Wu et al., 2025a), and AgiBot World (Bu et al., 2025a). These datasets paved the way for generalist policies, beginning with landmark models like RT-1 (Brohan et al., 2022) and RT-2 (Zitkovich et al., 2023), which established the paradigm of unifying large-scale vision-language pre-training with action generation. Subsequent initiatives like RT-X (O'Neill et al., 2024) and Octo (Team et al., 2024b) further consolidated data heterogeneity, while models such as PaLM-E (Driess et al., 2023) demonstrated the power of conditioning Large Language Models (LLMs) on high-fidelity visual inputs to enhance complex task planning and semantic grounding.

Beyond core training paradigm design, a significant research direction focuses on augmenting VLA models with richer world knowledge and diverse capabilities, including CrossFormer (Zhang & Yan, 2023), OpenVLA (Kim et al., 2024), HPT (Wang et al., 2024a), $\pi_0$ (Black et al., 2024), RDT (Liu et al., 2025b), TinyVLA (Wen et al., 2025b), GR00T (Bjorck et al., 2025), HybridVLA (Liu et al., 2025a), SwitchVLA (Li et al., 2025a), DTP (Fan et al., 2025), MLA (Liu et al., 2025c), and X-VLA (Zheng et al., 2025b). This typically involves leveraging representations pre-trained on vast internet-scale corpora. For instance, $\pi_{0.5}$ (Intelligence et al., 2025) and FSD (Yuan et al., 2025) integrate large-scale image-text pre-training to improve semantic understanding and visual grounding. Other works target higher-level cognitive abilities: CoT-VLA (Zhao et al., 2025) incorporates complex Chain-of-Thought (CoT) reasoning for planning, while InstructVLA (Yang et al., 2025) focuses on improving fidelity to natural language instructions. To strengthen the policy's grasp of the physical environment, SpatialVLA (Qu et al., 2025) emphasizes enhanced spatial reasoning. Furthermore, generative modeling techniques have been adapted, such as Diffusion-VLA (Wen et al., 2025c), which utilizes diffusion models for diverse action generation. Our framework, XR-1, deviates from traditional two-stage paradigms by introducing a novel three-stage process that synergizes human and robot data. Its key feature is an initial self-supervised stage for learning unified vision-motion representations. These representations act as auxiliary features to help pre-training and enhance data utilization for large-scale VLA models in the subsequent stage. This model-agnostic

approach ensures flexibility. We validate XR-1's flexibility by building upon base models like $\pi_0$ and SwitchVLA, yielding the high-performing XR-1 and efficient XR-1-Light models.

## 2.2 LATENT REPRESENTATION LEARNING

A major bottleneck in learning robust visuomotor policies from raw sensory inputs is the dimensional and semantic gap between high-fidelity pixel observations and low-dimensional motor commands. To effectively abstract away noise, high-dimensionality, and embodiment-specific details, prior work (Cui et al., 2023; Shafiullah et al., 2022; Lee et al., 2024; Zheng et al., 2025a; Xie et al., 2025) has extensively utilized latent representations as a critical intermediary layer between observations and actions. Current methodologies predominantly fall into two distinct, unimodal categories. The first category focuses on modeling the low-level motor dynamics by discretizing continuous robotic actions into a sequence of discrete latent tokens. This approach, exemplified by Behavior Transformers (BeT) (Shafiullah et al., 2022) and refined by methods such as QueST (Mete et al., 2024), transforms the continuous control problem into a tractable sequence-generation task. Recent works have further refined this direction. For instance, (Bauer et al., 2025) proposed a discrete latent action framework to enhance data efficiency, while ATE (Zhang et al., 2025) focused on effective feature alignment between visual input and the discretized action space. Similarly, approaches such as Moto (Chen et al., 2025) and Discrete Policy (Wu et al., 2025b) on discrete representations demonstrate the scalability of action tokenization for generalized control. However, these action-centric methods are fundamentally limited by their reliance on large volumes of high-quality, labeled robotic action data, which is time-consuming and expensive to acquire at the scale necessary for truly generalist agents.

In contrast, the second category seeks to learn representations by exploiting the vast abundance of unlabeled video data (Cui et al., 2023; Du et al., 2023; Hu et al., 2024; He et al., 2024; Ye et al., 2025; Cheang et al., 2025a), focusing on encoding generalized visual flow and state transitions observed in demonstrations. Models like C-BeT (Cui et al., 2023) and UniPi (Du et al., 2023) learn goal-conditioned behaviors from uncurated "play" data or text-conditioned video generation. More recent works, such as VPP (Hu et al., 2024), LAPA (Ye et al., 2025), and GR-2 (Cheang et al., 2025a), leverage large-scale actionless human videos and web videos for pre-training, aiming to capture generalized visual dynamics and task semantics. VPDD (He et al., 2024) leverages large-scale actionless human videos for pre-training and discrete diffusion modeling to enable effective robot policy learning with limited labeled data. Models like GO-1 (Bu et al., 2025a) leverage this video-centric approach to capture generalized, task-relevant visual dynamics. UVA (Li et al., 2025b) typically treats the vision and action modalities in isolation during the core representation learning phase.

However, by learning from action-free data, these methods lack explicit action grounding, creating a critical alignment gap between understanding visual change and executing the precise, fine-grained motor control required to effect that change. Crucially, by treating vision and action in isolation, existing unimodal paradigms fail to capture the causal link between observation and execution. We address this fundamental limitation with XR-1, a framework that introduces Unified Vision-Motion Codes (UVMC). UVMC is a novel, discrete latent representation learned jointly from both visual dynamics and robotic motion. By constructing a shared bimodal latent space, UVMC explicitly encodes the cause-and-effect relationship between seeing and acting, allowing it to abstract away embodiment-specific details while preserving core task semantics.

## 3 METHODOLOGY

### 3.1 OVERVIEW

Our goal is to build a versatile and generalist Vision-Language-Action (VLA) model that controls diverse robotic embodiments across tasks. At each inference step $t$, the policy $\pi$ receives a language instruction $l$ and multimodal observations $o = \langle c, m \rangle$, where $c \in \mathbb{R}^{K \times 3 \times H \times W}$ denotes $K$ RGB images from external or robot-mounted cameras, and $m$ represents proprioceptive states. The model then predicts the next action $\hat{a} = \pi(l, o)$ in terms of joint positions and gripper commands.

We introduce **XR-1**, a scalable framework for VLA learning across robots, tasks, and environments (Figure 2). Training proceeds in three stages. First, we learn a dual-branch VQ-VAE that encodes

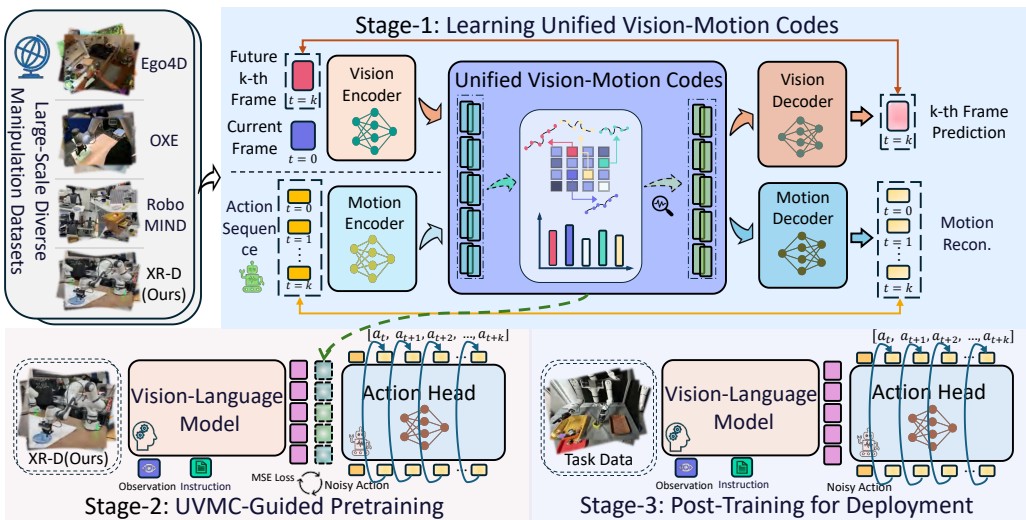

Figure 2: Overview of **X Robotic Model 1 (XR-1)**. In XR-1, we introduce the *Unified Vision-Motion Codes (UVMC)*, a discrete latent representation that jointly encodes visual dynamics and robotic motion. XR-1 adopts a three-stage training paradigm to enable precise low-level control across diverse robots and tasks.

visual dynamics and robot motion into a shared discrete latent space, and extract the *Unified Vision-Motion Codes (UVMC)*. In the second stage, these codes serve as supervision for large-scale pre-training of a policy on cross-embodiment datasets, enabling generalization across different robots and task distributions. Finally, the pretrained policy is fine-tuned on multi-task data collected from the target embodiment, which adapts the model to embodiment-specific dynamics and improves task success rates. This progressive design, including unified representation learning, cross-embodiment pretraining, and task-specific post-training, achieves both scalability and adaptability.

## 3.2 STAGE-1: LEARNING UNIFIED VISION-MOTION CODES

We design a dual-branch Vector Quantized Variational Autoencoder (VQ-VAE) (Van Den Oord et al., 2017) to learn the Unified Vision-Motion Codes (UVMC) in a self-supervised manner. Unlike prior works focusing solely on visual dynamics (Cui et al., 2023; Hu et al., 2024; Bu et al., 2025a; He et al., 2024; Ye et al., 2025; Cheang et al., 2025a; Du et al., 2023) or action sequences (Shafiullah et al., 2022; Wu et al., 2025b; Bauer et al., 2025; Zhang et al., 2025; Mete et al., 2024; Chen et al., 2025), our design explicitly unifies the two modalities in a discrete latent space and aligns them with an alighment regulrazation loss, providing complementary guidance for action prediction and enabling learning from heterogeneous sources such as human demonstrations.

**Visual Dynamic Code Extraction**. Vision captures universal dynamics across robots and environments. To encode temporal visual variations in the vision branch, we adopt an asymmetric VQ-VAE (Zhu et al., 2023c) structure tailored for future-frame prediction. Given two frames $c_t$ and $c_{t+h}$, the vision encoder $E_{\text{vis}}(\cdot)$ produces a latent code $z_{\text{vis}} = E_{\text{vis}}(c_t, c_{t+h})$, which compresses temporal changes over $h$ steps. The decoder then predicts the future frame via $\hat{c}_{t+h} = D_{\text{vis}}(c_t, z_{\text{vis}})$. Thus, $z_{\text{vis}}$ captures the essential visual dynamics.

**Robotic Motion Extraction**. The second branch encodes low-level actions and proprioceptive states. Specifically, the motion encoder $E_{\text{mo}}(\cdot)$ takes $(a_{t:t+h}, m_{t:t+h})$ as input and outputs $z_{\text{mo}} = E_{\text{mo}}(a_{t:t+h}, m_{t:t+h})$. Unlike the vision branch, no raw images or instructions are used here to ensure that the representation focuses purely on robotic dynamics. The motion decoder $D_{\text{mo}}(\cdot)$ then takes the latent motion embedding $z_{mo}$ and optional conditions $cd$ as input, such as the language instruction $l$, proprioceptive states $m$, and the observations $o$. The decoder reconstructs actions as $\hat{a}_{t:t+h} = D_{\text{mo}}(z_{mo}, cd)$. In our implementation, we use the proprioceptive states $m$ as the condition input.

**Unified Vision-Motion Codes**. To unify both modalities, we introduce a shared codebook $e \in \mathbb{R}^{d \times f}$ with $d$ discrete entries of dimension $f$. Encoder outputs $z_{\text{vis}}$ and $z_{\text{mo}}$ are quantized by nearest-

neighbor lookup: $z_{\text{vis}}^e = S(z_{\text{vis}}) = e_j$, where $j = \arg\min_i \|z_{\text{vis}} - e_i\|_2$, and $z_{\text{mo}}^e = S(z_{\text{mo}}) = e_j$, where $j = \arg\min_i \|z_{\text{mo}} - e_i\|_2$. Both decoders then condition on these quantized codes for reconstruction. Training follows standard VQ-VAE objectives (Van Den Oord et al., 2017), combining reconstruction losses with codebook and commitment regularization terms:

$$\mathcal{L}_{\text{vis}} = \|\hat{c}_{t+h} - c_{t+h}\|_1 + \beta\|sg(z_{\text{vis}}) - z_{\text{vis}}^e\|_2^2 + \beta\|z_{\text{vis}} - sg(z_{\text{vis}}^e)\|_2^2, \quad (1)$$

$$\mathcal{L}_{\text{mo}} = \|\hat{a}_{t:t+h} - a_{t:t+h}\|_1 + \beta\|sg(z_{\text{mo}}) - z_{\text{mo}}^e\|_2^2 + \beta\|z_{\text{mo}} - sg(z_{\text{mo}}^e)\|_2^2, \quad (2)$$

where $sg(\cdot)$ denotes stop-gradient. We set $\beta = 0.25$ in all experiments. To capture both the vision and motion signals, we concatenate the robotic motion codes $z_{\text{mo}}^e$ and visual dynamics codes $z_{\text{vis}}^e$ to obtain the Unified Vision-Motion Codes $z_{\text{uvmc}}^e$ for subsequent policy learning.

**Cross-Modality Alignment**. While motion codes provide precise control signals, visual embeddings may capture irrelevant factors (e.g., camera jitter). To mitigate this gap, we introduce an alignment loss that constrains visual codes to remain consistent with their motion counterparts:

$$\mathcal{L}_{\text{align}} = D_{\text{KL}}\left(q(z_{\text{mo}}) \,\|\, q(z_{\text{vis}})\right),$$

where $q(\cdot)$ denotes the posterior distribution in the codebook space. This grounding of perception in motor dynamics improves robustness and allows human-only demonstrations to be effectively mapped into the robot's action space.

**Final Training Objective**. The overall objective integrates reconstruction and alignment losses from different data sources. For robotic demonstrations, we jointly optimize $\mathcal{L}_{\text{total}}^{\text{robot}} = \mathcal{L}_{\text{vis}} + \mathcal{L}_{\text{mo}} + \mathcal{L}_{\text{align}}$, where $\mathcal{L}_{\text{vis}}$ and $\mathcal{L}_{\text{mo}}$ are the VQ-VAE losses for visual and motion branches, and $\mathcal{L}_{\text{align}}$ enforces cross-modal consistency. For human demonstrations, where low-level actions are unavailable, the objective naturally reduces to $\mathcal{L}_{\text{total}}^{\text{human}} = \mathcal{L}_{\text{vis}}$. This design allows training on both robot rollouts and purely visual human data. Further architectural details are provided in Appendix 6.2.

### 3.3 STAGE-2: UVMC-GUIDED PRETRAINING FOR GENERALIST POLICY

After learning the Unified Vision-Motion Codes (UVMC) with the dual-branch VQ-VAE, we integrate it into policy learning to enhance low-level control. The policy $\pi(\cdot)$ follows a standard VLA design with a VLM $F(\cdot)$ and an action head $H(\cdot)$. Learnable tokens $t$ are introduced into the VLM input, enabling $F(\cdot)$ to predict the UVMC. The prediction loss is defined as $\mathcal{L}_{\text{uvmc}} = \|F(l, o, t) - z_{\text{uvmc}}^e\|_2^2$. In parallel, the action head is pretrained on robot datasets using an action loss $\mathcal{L}_{\text{act}}$, which may be generative or autoregressive depending on the model variant. The overall objective is $\mathcal{L} = \mathcal{L}_{\text{uvmc}} + \mathcal{L}_{\text{act}}$. This joint training encourages the backbone to internalize structured vision-motion representations while ensuring effective large-scale action pretraining.

### 3.4 STAGE-3: POST-TRAINING FOR DEPLOYMENT

After large-scale UVMC-guided pretraining, the model acquires strong abilities in extracting unified vision-motion knowledge and producing foundation-level actions. To further improve performance on downstream control tasks, we introduce a post-training stage where the VLA policy is fine-tuned with task-specific datasets using an action loss $\mathcal{L}_{\text{act}}$. A key advantage of our framework is its model-agnostic design: it can be directly applied to different VLA architectures. This flexibility enables users to integrate diverse backbones while consistently benefiting from our framework.

### 3.5 DATA COLLECTION AND IMPLEMENTATION DETAILS

**Dataset Collection**. To support large-scale pretraining, we curate a comprehensive dataset by integrating four complementary sources: Open-X (O'Neill et al., 2024), RoboMIND (Wu et al., 2025a), Ego4D (first-person human activity videos) (Grauman et al., 2022), and XR-D (our in-house collection spanning multiple robot embodiments).

Table 1 summarizes the distribution of episodes and frames across these datasets, together with their relative proportions. Since the number of episodes and frames varies significantly

Table 1: Dataset Statistics.

| Dataset | Episodes | Frames | Weight |
|---------|----------|--------|--------|
| OXE | 978k | 59.3M | 40% |
| RoboMIND | 69k | 21.4M | 15% |
| XR-D | 158k | 69.1M | 35% |
| Ego4D | 59k | 14.3M | 10% |

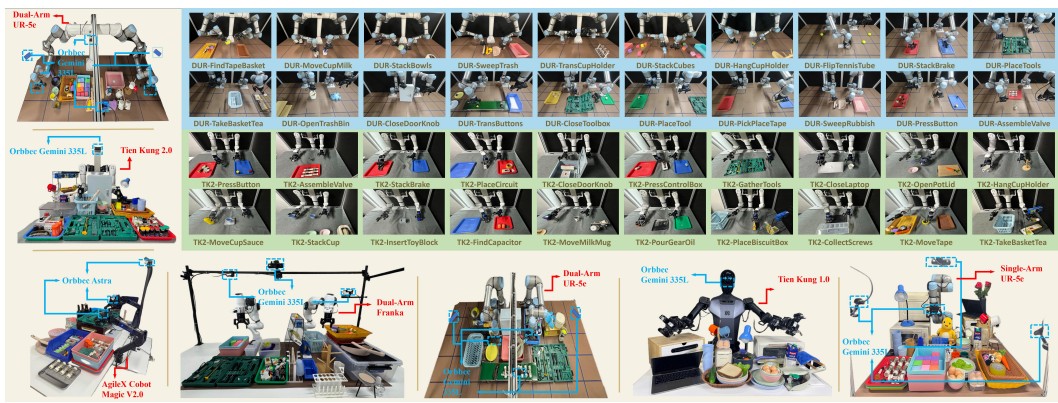

Figure 3: Experimental Setup. We evaluate XR-1 across six robot embodiments (Tien Kung 1.0/2.0, Single-/Dual-Arm UR-5e, Dual-Arm Franka, and AgileX Cobot Magic 2.0), covering more than 120 manipulation tasks with over 14k rollouts.

among different sources, we assign dataset-specific sampling weights during training to balance contributions and prevent overfitting to dominant datasets. We provide more details of the datasets in Appendix 6.3.

**Implementation Details**. The framework is model-agnostic. Our main instantiation adopts the design of $\pi_0$ (Black et al., 2024), which is built on PaliGemma (Beyer et al., 2024) (SigLIP visual encoder (Zhai et al., 2023) + Gemma backbone (Team et al., 2024a) + action head), while a lightweight variant (XR-1-Light) built up on SwitchVLA (Li et al., 2025a) uses Florence-2 (Xiao et al., 2024) to reduce computation cost with minimal performance drop.

## 4 EXPERIMENTS

We evaluate XR-1 through four key questions: (1) How does it compare with state-of-the-art (SOTA) vision-language-action (VLA) models? (2) Does large-scale pretraining endow the model with fundamental execution skills and rapid adaptation? (3) How well does it generalize to novel objects, background shifts, distractors, and lighting variations? (4) What is the impact of different components and training strategies on performance? To address these questions, we conduct extensive real-world evaluations on over 120 tasks across six robotic embodiments. The tasks cover diverse and challenging scenarios, including bimanual collaboration, dexterous manipulation, deformable object handling, contact-rich interactions, dynamic environments, and long-horizon manipulation. We benchmark our approach against multiple strong VLA baselines.

### 4.1 EXPERIMENT SETUP

**Real-World Robotic Setup**. We evaluate XR-1 on six heterogeneous robotic embodiments (Figure 3): Tien Kung 1.0/2.0, Single-/Dual-Arm UR-5e, Dual-Arm Franka, and AgileX Cobot Magic 2.0. All robots are equipped with parallel grippers and multiple cameras from complementary viewpoints. For each robotic embodiment, we design 20 tasks and collect expert demonstrations via teleoperation, recording synchronized multi-view RGB streams and proprioceptive states (e.g., joint positions and gripper commands). The 20 task examples for Dual-Arm UR-5e and Tien Kung 2.0 are shown in Figure 3, while full task details are provided in the Appendix 6.11.

**Training and Evaluation Protocol.** We adopt a three-stage training pipeline. First, XR-1 is pretrained on large-scale heterogeneous datasets (RoboMIND (Wu et al., 2025a), Open-X (O'Neill et al., 2024), XR-D, Ego4D (Grauman et al., 2022)), enabling the dual-branch VQ-VAE to learn the Unified Vision-Motion Codes (UVMC). Second, we pretrain the policy on XR-D to integrate cross-embodiment knowledge. Finally, the policy is fine-tuned on data of specified tasks. For evaluation, we conduct 20 rollouts per task, with human evaluators determining success based on goal completion. Final performance is reported as the success rate.

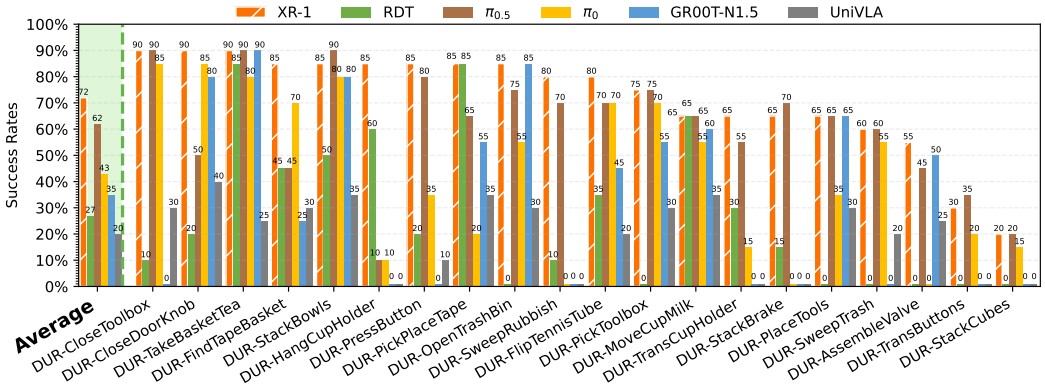

Figure 4: Success rate results across 20 tasks on Dual-Arm UR-5e.

Table 2: Success rate results across 20 tasks on Tien Kung 2.0.

| Method | TK2-Press Button | TK2-Assemble Valve | TK2-Stack Brake | TK2-Place Circuit | TK2-Press ControlBox | TK2-Close DoorKnob | TK2-Gather Tools | TK2-Close Laptop | TK2-Open PotLid | TK2-Hang CupHolder | - |
|---|---|---|---|---|---|---|---|---|---|---|---|
| UniVLA | 25 | 0 | 25 | 50 | 10 | 0 | 0 | 20 | 35 | 0 | - |
| RDT | 15 | 0 | 0 | 65 | 20 | 0 | 0 | 0 | 90 | 0 | - |
| GR00T-N1.5 | 85 | 20 | 85 | 90 | 0 | 0 | 20 | 0 | 75 | 0 | - |
| $\pi_0$ | 85 | 10 | 55 | 70 | 85 | 0 | 20 | 85 | 85 | 20 | - |
| $\pi_{0.5}$ | 80 | 0 | 65 | 60 | 40 | 25 | 20 | 90 | 80 | 35 | - |
| XR-1 (ours) | 90 | 15 | 90 | 90 | 90 | 85 | 25 | 90 | 85 | 75 | - |

| | TK2-Move CupSauce | TK2-Stack Cup | TK2-Insert ToyBlock | TK2-Find Capacitor | TK2-Move MilkMug | TK2-Pour GearOil | TK2-Place BiscuitBox | TK2-Collect Screws | TK2-Move Tape | TK2-Take BasketTea | Avg. |
|---|---|---|---|---|---|---|---|---|---|---|---|
| UniVLA | 45 | 30 | 0 | 25 | 35 | 35 | 0 | 0 | 20 | 0 | 17.8 |
| RDT | 50 | 25 | 0 | 0 | 0 | 75 | 0 | 0 | 0 | 0 | 17.0 |
| GR00T-N1.5 | 50 | 55 | 0 | 60 | 55 | 70 | 25 | 0 | 70 | 0 | 38.0 |
| $\pi_0$ | 60 | 20 | 10 | 0 | 80 | 55 | 0 | 0 | 0 | 75 | 40.8 |
| $\pi_{0.5}$ | 70 | 25 | 20 | 0 | 75 | 75 | 0 | 0 | 0 | 60 | 41.0 |
| XR-1 (ours) | 70 | 85 | 55 | 75 | 90 | 85 | 75 | 15 | 70 | 85 | 72.0 |

## 4.2 RESULTS ON REAL-WORLD ROBOTIC TASKS

**Baseline Methods**. We compare XR-1 with strong VLA models, including $\pi_{0.5}$ (Intelligence et al., 2025), $\pi_0$ (Black et al., 2024), RDT (Liu et al., 2025b), UniVLA (Bu et al., 2025b), and GR00T-N1.5 (Bjorck et al., 2025). We note a performance degradation with the Lerobot implementation of $\pi_0$. The results of $\pi_0$ reported in this paper are based on the original JAX implementation.

**Results on Dual-Arm UR-5e**. Figure 4 reports success rates across 20 tasks on the Dual-Arm UR-5e. XR-1 surpasses all baselines by a large margin. For instance, in *DUR-FindTapeBasket*, it achieves 85% success compared to 50% from $\pi_0$. Several baselines even collapse to 0% performance on harder tasks, which we attribute to insufficient auxiliary supervision and gradient conflicts during multi-task optimization. In contrast, XR-1 leverages UVMC for richer training signals, yielding more robust representations and stable optimization across diverse objectives. The corresponding tabular results can be found in Appendix Table 13.

**Results on Tien Kung 2.0**. We further evaluate transferability on Tien Kung 2.0 over another 20 tasks in Table 2. Unlike the UR-5e, this robot is *unseen during pretraining* (e.g., Stages 1 and 2 for XR-1), making the evaluation a stringent embodiment-transfer benchmark. Despite this challenge, XR-1 again outperforms all baselines; e.g., in *TK2-MoveCupSauce*, it reaches 70% versus 60% for $\pi_0$. These results indicate that UVMC effectively encodes embodiment-agnostic dynamics into a shared latent space, enabling efficient transfer of prior knowledge to novel robotic platforms.

**Results on Other Robots**. XR-1 consistently outperforms all other methods across four diverse robotic arm configurations, achieving a significant relative gain over the strongest baseline. Additional experimental results for Tien Kung 1.0 in Table 9, Dual-Arm Franka in Table 10, AgileX Cobot Magic V2.0 in Table 11, and Single-Arm UR-5e in Table 12 are provided in Appendix 6.4.

## 4.3 GENERALIZATION ANALYSIS

**Out-of-Box Evaluation**. We assess the foundation ability of XR-1 after Stage-1 and Stage-2, without any post-training in Stage-3. We evaluate on 7 tasks each from the Dual-Arm UR-5e and Dual-Arm Franka in XR-D, covering only 0.9% of the XR-D dataset. For fair comparison, baselines without XR-D pretraining are fine-tuned on data from these tasks before evaluation. As shown in

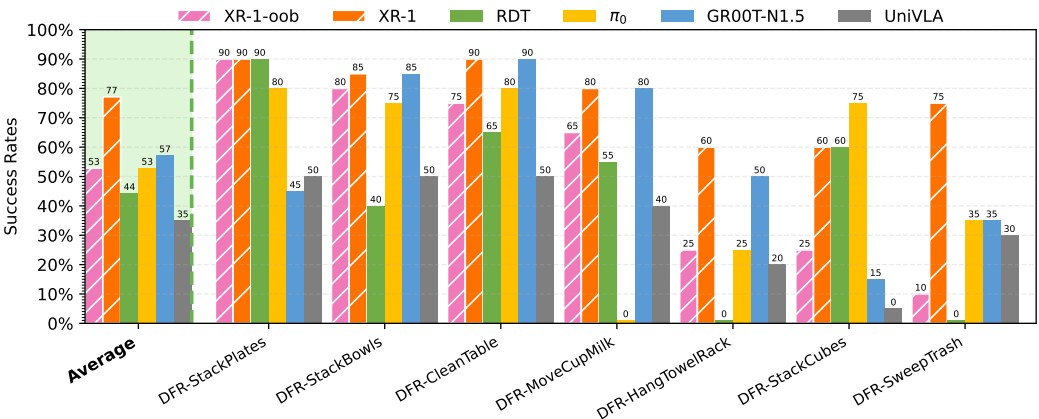

Figure 5: Out-of-box evaluation results of 7 tasks on Dual-Arm Franka.

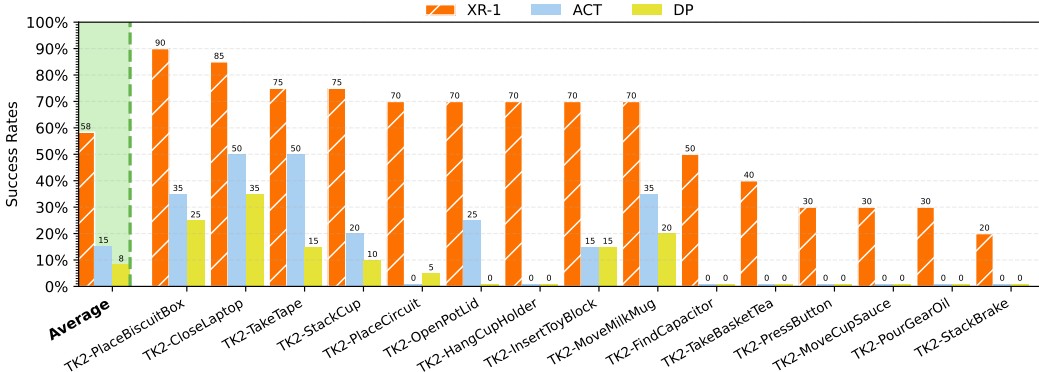

Figure 6: Fast adaptation on Tien Kung 2.0. Tien Kung 2.0 is an unseen embodiment in XR-D. In this setup, XR-1 adapts to 15 novel tasks with one model using only 20-shot demonstrations per task, while baselines (ACT and DP) are trained per task.

Figure 5, the pretrained XR-1-oob model, even without adaptation, achieves performance close to GR00T-N1.5 while outperforming RDT and UniVLA. *This robustness stems from UVMC learning, which aligns multimodal dynamics across embodiments into a unified latent space, thereby enabling strong generalization with extremely limited task-specific supervision.* Additional results on Dual-Arm UR-5e are provided in Appendix 6.5

**Fast Adaptation to New Tasks**. We further evaluate whether XR-1 can rapidly adapt to unseen tasks with limited demonstrations. Specifically, we collect 15 new tasks on both the Dual-Arm UR-5e and Tien Kung 2.0 (unseen in XR-D), each with 20 trajectories. XR-1 is trained jointly across these tasks, while single-task baselines, ACT (Zhao et al., 2023) and Diffusion Policy (DP) (Chi et al., 2023), are trained independently per task. As shown in Figure 6, XR-1 achieves significantly higher success rates than ACT and DP, despite the evaluation setting favoring the baselines. *This advantage stems from large-scale pretraining combined with UVMC supervision, enabling XR-1 to extract transferable features from few-shot data and adapt effectively across diverse embodiments.* Additional results on Dual-Arm UR-5e are provided in Appendix 6.5.

**Generalization to Unseen Scenrios**. We further evaluate XR-1 on unseen conditions to assess its out-of-distribution generalization. As shown in Figure 7, we test on (*i*) novel objects (e.g., unseen rubbish or dustpans), (*ii*) dynamic and static distractors, (*iii*) illumination changes, and (*iv*) background variations. As shown in Table 3, XR-1 consistently outperforms the strong VLA method $\pi_0$ across all settings. It demonstrates clear gains on novel objects, improved robustness under distractor interference, and stable performance when background and lighting variations are introduced. *These results highlight XR-1's strong generalization not only across embodiments and tasks but also under diverse environmental shifts never encountered during pretraining or fine-tuning, underscoring its potential for real-world deployment.*

Figure 7: Unseen scenario task setup on Dual-Arm Franka.

Table 3: Generalization results of XR-1 on unseen scenarios.

| Method | DFR-SweepTrash | | | DFR-HangCup | | |
|---|---|---|---|---|---|---|
| | Novel Objects (rubbish) | Novel Objects (dustpan) | Dynamic Distractors | Background Variations | Illumination Changes | Static Distractors |
| $\pi_0$ | 15 | 50 | 5 | 30 | 15 | 10 |
| XR-1 (ours) | 65 | 60 | 55 | 55 | 30 | 30 |

Table 4: Ablation study of XR-1. In Stage-1 and Stage-2, "DT" indicates training directly on the downstream task data.

| Exp. | Instantiation | Stage-1 | Stage-2 | Stage-3 | DUR-Clean Table | DUR-Find TapeBasket | DUR-Move CupMilk | DUR-Stack Bowls | DUR-Sweep Trash | DUR-Trans CupHolder | Avg |
|---|---|---|---|---|---|---|---|---|---|---|---|
| 1 | XR-1-Light | $\times$ | $\times$ | $\checkmark$ | 0 | 70 | 0 | 75 | 60 | 50 | 42.5 |
| 2 | XR-1-Light | DT | DT | $\checkmark$ | 40 | 90 | 10 | 90 | 60 | 55 | 57.5 |
| 3 | XR-1 | $\times$ | $\times$ | $\checkmark$ | 0 | 50 | 20 | 55 | 0 | 45 | 28.3 |
| 4 | XR-1 w/o KL | DT | DT | $\checkmark$ | 45 | 55 | 35 | 60 | 30 | 65 | 48.3 |
| 5 | XR-1 | DT | DT | $\checkmark$ | 50 | 75 | 65 | 80 | 60 | 70 | 66.7 |
| 6 | XR-1 | 1% | DT | $\checkmark$ | 15 | 60 | 10 | 55 | 15 | 20 | 29.2 |
| 7 | XR-1 | 10% | DT | $\checkmark$ | 25 | 60 | 25 | 60 | 20 | 40 | 38.3 |
| 8 | XR-1 | 50% | DT | $\checkmark$ | 25 | 80 | 65 | 80 | 20 | 50 | 53.3 |
| 9 | XR-1 | 100% | DT | $\checkmark$ | 60 | 80 | 70 | 85 | 40 | 55 | 65.0 |
| 10 | XR-1 | 100% | XR-D | $\checkmark$ | 70 | 85 | 80 | 90 | 85 | 80 | 81.6 |

## 4.4 ABLATION STUDY

To disentangle the contribution of each component in XR-1, we conduct ablations on six manipulation tasks using the Dual-Arm UR-5e. Table 4 summarizes success rates under different configurations, covering model capacity, UVMC learning, cross-modal alignment, and dataset scaling. Due to space limitation, additional experimental results are provided in Appendix 6.6.

**Lightweight Models.** We first evaluate a compact variant, XR-1-Light, with only 230MB trainable parameters. Comparing Exp. 1 and Exp. 2 shows that incorporating UVMC with downstream data improves average success from 42.5% to 57.5%. This indicates that UVMC provides substantial benefits even for low-capacity models trained on limited data.

**UVMC and Cross-Modal Alignment.** Exps. 3–5 examine the role of UVMC together with a cross-modal alignment loss between vision and motion. Performance consistently improves as these components are added, confirming their complementary importance for feature learning across tasks.

**Scaling with Pretraining Data.** Exps. 6–9 vary the scale of Stage-1 pretraining data from 1% to 100%. Results show a clear monotonic gain in success rates as more data is used, highlighting the central role of large-scale pretraining for robust generalization.

## 5 CONCLUSION

We presented **X Robotic Model 1 (XR-1)**, a unified framework for versatile and scalable vision-language-action learning that addresses the key limitations of existing approaches: precise low-level action generation and cross-domain multimodal knowledge exploitation across heterogeneous data sources. Central to our approach is the *Unified Vision-Motion Codes (UVMC)*, which serve as embodiment-agnostic abstractions aligning visual dynamics with motor control through a shared discrete latent space. By utilizing a three-stage training paradigm, XR-1 achieves robust performance across diverse robots and tasks while significantly outperforming state-of-the-art baselines such as $\pi_{0.5}$, $\pi_0$, RDT, UniVLA, and GR00T-N1.5. Our results highlight the importance of multimodal alignment for embodied AI and suggest promising directions toward general-purpose robotic agents capable of interacting with the physical world and adapting seamlessly to new environments.

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

# 6 APPENDIX

## 6.1 THE USE OF LARGE LANGUAGE MODELS (LLMs)

In preparing this manuscript, we employed Large Language Models (LLMs) solely for assistance in academic writing, including text refinement and polishing. No other use of LLMs was involved in the research process, data analysis, or experimental design. All conceptual development, algorithmic contributions, and empirical evaluations were conducted independently by the authors.

## 6.2 IMPLEMENTATION DETAILS

In this section, we provide a detailed description of the XR-1 framework, focusing on the architecture and training of the dual-branch VQ-VAE. The model is designed to encode both vision dynamics and robotic motion into a shared discrete latent space, thereby enabling seamless integration of perception and control.

### 6.2.1 DUAL-BRANCH VQ-VAE

To achieve a unified latent representation, we introduce a dual-branch VQ-VAE consisting of two complementary encoders, a vision encoder and a motion encoder, that map their respective modalities into a common discrete codebook. Each branch is paired with a decoder to facilitate reconstruction during pretraining. The overall design ensures that the majority of representational capacity resides in the encoders, while the decoders primarily serve as auxiliary components for reconstruction.

**Vision Branch.** The vision branch processes raw image observations $\{o_t, o_{t+h}\}$ and encodes them into compact latent tokens.

**Vision Branch Encoder.** We adopt SigLIP (Zhai et al., 2023) as the backbone vision encoder, comprising approximately 400M parameters. This encoder extracts high-level features from visual inputs. To capture temporal dynamics beyond static representations, we incorporate a visual dynamic module inspired by (Chen et al., 2025). This module is implemented as a four-layer transformer (ViT (He et al., 2022)) with 32M parameters, which compresses vision dynamic information into a fixed number of latent tokens by querying dynamic features.

**Vision Branch Decoder.** For reconstruction, we employ a ViT-based decoder with 12 transformer layers (94M parameters). Importantly, the encoder contains roughly five times more parameters than the decoder. This asymmetry is intentional: by allocating more capacity to encoding, we encourage the model to produce informative latent tokens that simplify downstream decoding. Consequently, the decoder remains lightweight since its role is auxiliary rather than representationally dominant.

During training, all parameters in both the SigLIP backbone and the dynamic module remain fully trainable. Additional details regarding training hyperparameters are provided in Table 5.

**Motion Branch.** The motion branch encodes action sequences $\{a_{t:t+h}\}$ into discrete motion codes.

**Motion Branch Encoder.** To capture temporal dependencies across actions, we employ 1D causal strided convolutions (Van Den Oord et al., 2016), which progressively reduce sequence length $h$ while preserving causality. The stride configuration determines the degree of temporal abstraction achieved at each stage. Following this convolutional compression, an 8-layer transformer encoder (34M parameters) further contextualizes action embeddings before quantization into discrete tokens.

**Motion Branch Decoder.** For action reconstruction, we leverage Gemma (Beyer et al., 2024), an autoregressive language model with approximately 300M parameters. The design closely follows the action expert structure in $\pi_0$ (Black et al., 2024), integrating diffusion-based supervision for reconstructing low-level actions from motion codes. Pretraining this decoder equips it with strong generative priors over action sequences, thereby providing an effective initialization for downstream policy learning. Additional details regarding training hyperparameters are provided in Table 5

Overall, this dual-branch architecture ensures that both perception and motion are represented in a unified tokenized space via vector quantization (VQ), enabling scalable pretraining across multi-modal data sources.

Table 5: Implementation Details of Dual-Branch VQ-VAE.

| | Hyperparameter | Value | | Hyperparameter | Type | Params. |
|---|---|---|---|---|---|---|
| | Batch Size | 960 | | Vision Encoder | SigLIP | 400M |
| | Learning Rate | 1e-4 | | Vision Dynamic Encoder | ViT | 32M |
| | Optimizer | AdamW | | Vision Decoder | ViT | 94M |
| | Trainable Parameters | 0.9B | | Vision Recons. Loss | MSE | - |
| Hyper-parameter | Motion/Vision Codebook Category | 256 | Network Architectures | Action Encoder | Convolution and Transformer | 33M |
| | Motion/Vision Codebook Embed. Dim | 256 | | Action Decoder | Transformer Decoder | 300M |
| | Motion/Vision Code Num. | 13 | | Action Recons. Loss | Flow Matching | - |
| | Action Sequence | 50 | | - | - | |
| | Vision Interval | 50 | | - | - | |
| | Training Step | 275K | | - | - | |

### 6.2.2 XR-1 Models

**XR-1.** The proposed framework is designed to be model-agnostic, making it compatible with a wide range of vision-language-action (VLA) architectures. In this work, we instantiate XR-1 using a configuration inspired by the baseline policy $\pi_0$ (Black et al., 2024) while introducing several key modifications that enable more structured representation learning. Specifically, XR-1 builds upon the PaliGemma architecture (Beyer et al., 2024), which integrates a SigLIP-based visual encoder (Zhai et al., 2023) with approximately 400 million parameters and a Gemma transformer backbone (Team et al., 2024a) with an action prediction head containing around 2.6 billion parameters. This design largely mirrors $\pi_0$ in terms of scale and backbone selection, but diverges in how supervision is introduced.

Instead of directly optimizing for action prediction as in $\pi_0$, XR-1 leverages the UVMC produced by a Dual-Branch VQ-VAE as intermediate supervisory signals. The joint representation $z_{uvmc}$ encodes both motion and visual dynamics information, which serves as guidance for training. To incorporate this signal effectively, we introduce two learnable tokens, $[ZMO]$ and $[ZVIS]$, that are responsible for predicting the robotic motion codes and the visual dynamics codes. These predictions are optimized using mean squared error loss against their respective targets. By enforcing this disentangled supervision on both motor control and perceptual dynamics, XR-1 encourages stronger alignment between perception and action.

To ensure fairness in evaluation, XR-1 is initialized from PaliGemma's publicly available pretrained checkpoint rather than directly adopting the released weights of $\pi_0$. This avoids potential confounding effects due to differences in pretraining objectives or data exposure. Overall, XR-1 extends beyond $\pi_0$ by introducing structured supervision through VQ-VAE latent codes and dedicated learnable tokens for motion and visual prediction, while maintaining compatibility with large-scale pretrained models such as PaliGemma. Additional details regarding training hyperparameters are provided in Table 6.

**XR-1-Light.** To further highlight the flexibility of our approach, we introduce **XR-1-Light**, a lightweight variant of XR-1 that significantly reduces computational cost while maintaining com-

Table 6: Implementation Details of XR-1.

| | Hyperparameter | Value | | Hyperparameter | Value |
|---|---|---|---|---|---|
| Hyper-parameter | Batch size | 640 | Network Architectures | Decoder layer | 18 |
| | Learning rate | 1e-4 | | Transformer hidden dim | 2048 |
| | Optimizer | AdamW | | Heads num | 8 |
| | $[ZMO]$ Number | 13 | | Action Decoder layer | 18 |
| | $[ZVIS]$ Number | $13*view_{num}$ | | Action Transformer hidden dim | 1024 |
| | Action sequence | 50 | | Action Heads num | 8 |
| | Training step | 300k | | Action loss | flow matching |

petitive performance. The motivation behind XR-1-Light is to replace the large-scale PaliGemma backbone, which contains nearly 3 billion parameters, with a more efficient vision-language model (VLM) without sacrificing the ability to capture rich multimodal representations. For this purpose, we adopt Florence-2 (Xiao et al., 2024), a transformer-based model with approximately 230 million parameters, as the backbone within the SwitchVLA framework (Li et al., 2025a). This substitution enables faster training and inference while lowering memory requirements, making XR-1-Light more suitable for resource-constrained scenarios.

Despite its reduced scale, XR-1-Light preserves the core design principles of XR-1. In particular, it continues to leverage the supervisory signal UVMC from the Dual-Branch VQ-VAE, which encodes both robotic motion and visual dynamics. To integrate this supervision effectively, we employ two learnable tokens, $[ZMO]$ and $[ZVIS]$, that are responsible for predicting the motion codes and the visual dynamics codes. Unlike in XR-1 where these tokens are attached to a decoder-only transformer backbone, in Florence-2 they are inserted between the encoder and decoder layers. This design allows the encoder to specialize in extracting structured latent representations aligned with UVMC, while enabling the decoder to function as an action expert that generates task-specific predictions conditioned on these learned codes.

A notable difference between XR-1 and XR-1-Light lies in their training strategies. While XR-1 benefits from pretraining on XR-D before fine-tuning on downstream tasks, XR-1-Light omits this stage due to its lightweight architecture. Instead, it is directly fine-tuned on task-specific datasets. This choice reflects a trade-off: although pretraining could potentially enhance generalization, direct fine-tuning allows us to fully exploit Florence-2's efficiency without incurring additional computational overhead.

In summary, XR-1-Light demonstrates that our framework can be instantiated not only with large-scale backbones such as PaliGemma but also with compact VLMs like Florence-2. By maintaining structured supervision through $z_{uvmc}$ while reducing parameter count by more than an order of magnitude, XR-1-Light provides a practical alternative that balances performance with efficiency. Additional details regarding training hyperparameters are provided in Table 7.

Table 7: Implementation Details of XR-1-Light

| | Hyperparameter | Value | | Hyperparameter | Value |
|---|---|---|---|---|---|
| Hyper-parameter | Batch Size | 160 | Network Architectures | Encoder Layer | 6 |
| | Learning Rate | 5e-5 | | Transformer Hidden Dim. | 768 |
| | Optimizer | AdamW | | Heads Num. | 12 |
| | $[ZMO]$ Number | 13 | | Action Decoder Layer | 6 |
| | $[ZVIS]$ Number | $13*view_{num}$ | | Action Transformer Hidden Dim. | 768 |
| | Action Sequence | 50 | | Action Heads Num. | 12 |
| | Training Step | 50K | | Action Loss | Flow Matching |

### 6.2.3 TRAINING AND INFERENCE

The training of our framework is organized into three stages: UVMC learning, UVMC-guided pre-training, and policy fine-tuning. Each stage progressively aligns perception, representation, and control while balancing computational efficiency.

In the first stage, the UVMC module, containing approximately 0.9B parameters, is pretrained on large-scale multimodal data. This process consumed roughly 38,400 GPU hours on a cluster of 80

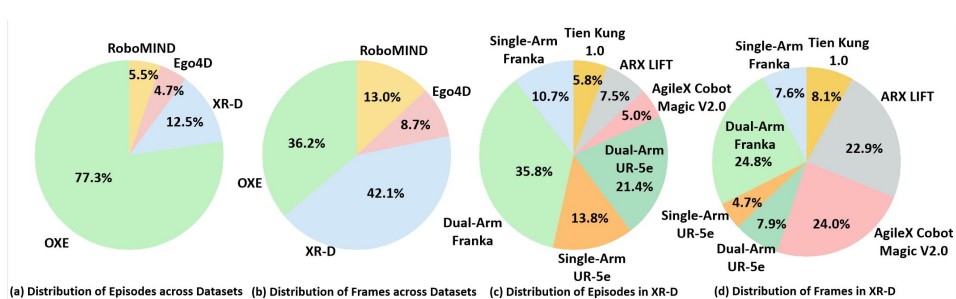

Figure 8: Overview of the pretraining datasets used for XR-1. We combine Open-X, RoboMIND, Ego4D, and our dataset XR-D, with a total of ∼1,264k episodes and 110M frames.

NVIDIA A100 GPUs (80GB each), enabling the model to capture both motion and visual dynamics representations.

The second stage involves policy pretraining, where the complete model scales up to about 4B parameters. This step also required around 38,400 GPU hours on the same hardware configuration. The objective here is to integrate the pretrained UVMC representations into a unified vision-language-action policy.

In the final stage, policy fine-tuning is performed for embodiment-specific adaptation. Each embodiment configuration is fine-tuned across 20 downstream tasks using 8 A100 GPUs (80GB), requiring approximately 576 GPU hours per embodiment. This ensures that XR-1 and its variants generalize effectively to diverse robotic environments while remaining computationally practical.

For inference, we emphasize both responsiveness and throughput. The system operates with an action chunk inference frequency of about 5 Hz while maintaining an average action-level inference rate close to 200 Hz (actions per second). These frequencies are achieved on a single commercially available RTX 4090 GPU (24GB), demonstrating that despite large-scale pretraining costs, deployment remains efficient without reliance on massive compute resources.

## 6.3 DATASET CURATION

Large-scale pretraining has consistently been shown to enhance both generalization and rapid adaptation in multimodal learning systems. Motivated by these findings, we curate a comprehensive dataset tailored for robotic manipulation, integrating diverse sources of visual, linguistic, and action-centric data. Our dataset construction draws from four complementary resources: *Open-X* (O'Neill et al., 2024), which provides large-scale open-world manipulation trajectories; *RoboMIND* (Wu et al., 2025a), a benchmark emphasizing reasoning-driven robotic tasks; *Ego4D* (Grauman et al., 2022), a first-person human activity dataset offering rich egocentric perspectives; and *XR-D*, our in-house collection spanning multiple robotic embodiments and task domains. Together, these sources cover a wide spectrum of sensory modalities, embodiment variations, and task complexities, forming a foundation for scalable pretraining.

The training procedure is organized into three progressive stages. In **Stage-1**, we pretrain a dual-branch VQ-VAE on the combined datasets to learn disentangled latent representations of motion and visual dynamics. In **Stage-2**, we leverage XR-D to pretrain the vision-language-action (VLA) backbone, aligning multimodal perception with action generation across diverse embodiments. Finally, in **Stage-3**, we fine-tune on novel scenes and previously unseen tasks outside XR-D in order to rigorously assess transferability and generalization beyond the pretraining distribution.

A detailed breakdown of dataset statistics, including scale, modality coverage, and embodiment diversity across all sources used for UVMC pretraining, is provided in Figure 8 and Table 8.

Table 8: Pretraining Dataset Details.

| Dataset | Episode | Frames | Weight |
|---|---|---|---|
| **OXE (O'Neill et al., 2024)** | **978,582** | **59.3M** | **40%** |
| FMB Dataset (Luo et al., 2025) | 8611 | 1137340 | 0.88% |
| DROID (Khazatsky et al., 2024) | 92233 | 27044326 | 9.43% |
| Language Table (Lynch et al., 2023) | 442226 | 7045476 | 45.19% |
| Berkeley Autolab UR5 (Chen et al.) | 896 | 87783 | 0.09% |
| Berkeley Fanuc Manipulation (Zhu et al., 2023a) | 415 | 62613 | 0.04% |
| Berkeley Cable Routing (Luo et al., 2024) | 1482 | 38240 | 0.15% |
| Berkeley Gnm Cory Hall (Kahn et al., 2018) | 7331 | 156012 | 0.75% |
| Berkeley Gnm Recon (Shah et al., 2021) | 11834 | 610907 | 1.21% |
| Berkeley Gnm Sac Son (Hirose et al., 2023) | 2955 | 241059 | 0.30% |
| Berkeley MVP (Radosavovic et al., 2023b) | 480 | 45308 | 0.05% |
| Berkeley RPT (Radosavovic et al., 2023a) | 908 | 392578 | 0.10% |
| Bridge (Ebert et al., 2022; Walke et al., 2023) | 25460 | 813372 | 2.60% |
| BC-Z (Jang et al., 2022) | 43264 | 6015535 | 4.42% |
| Taco Play (Rosete-Beas et al., 2023; Mees et al., 2023) | 3242 | 213972 | 0.33% |
| NYU Franka Play Dataset (Cui et al., 2023) | 365 | 34448 | 0.04% |
| Asu Table Top (Zhou et al., 2022b; 2023) | 110 | 26113 | 0.01% |
| Austin Buds Dataset (Zhu et al., 2022) | 50 | 34112 | 0.01% |
| Austin Sailor Dataset (Nasiriany et al., 2022) | 240 | 353094 | 0.02% |
| Austin Sirius Dataset (Liu et al., 2022) | 559 | 279939 | 0.05% |
| CMU Play Fusion (Chen et al., 2023) | 576 | 235922 | 0.05% |
| CMU Stretch (Bahl et al., 2023; Mendonca et al., 2023) | 135 | 25016 | 0.01% |
| Columbia Cairlab Pusht Real (Chi et al., 2023) | 122 | 24924 | 0.01% |
| DLR EDAN Shared Control (Vogel et al., 2020; Quere et al., 2020) | 104 | 8928 | 0.01% |
| DLR Sara Grid Clamp (Padalkar et al., 2023a) | 107 | 7622 | 0.01% |
| DLR Sara Pour (Padalkar et al., 2023b) | 100 | 12971 | 0.01% |
| DobbE (Shafiullah et al., 2023) | 5208 | 1139911 | 0.52% |
| Stanford Hydra Dataset (Belkhale et al., 2023) | 570 | 358234 | 0.06% |
| Tokyo U Lsmo (Osa, 2022) | 50 | 11925 | 0.01% |
| Toto (Zhou et al., 2022a) | 902 | 294139 | 0.10% |
| UCSD Kitchen Dataset (Yan & Wang, 2023) | 150 | 3970 | 0.02% |
| UCSD Pick and Place Dataset (Feng et al., 2023) | 1355 | 67750 | 0.14% |
| UTAustin Mutex (Shah et al., 2023) | 1500 | 361883 | 0.15% |
| U-Tokyo PR2 Opening Fridge (Oh et al., 2023) | 64 | 9140 | 0.01% |
| U-Tokyo PR2 Tabletop Manipulation (Oh et al., 2023) | 192 | 26346 | 0.02% |
| U-Tokyo xArm Bimanual (Matsushima et al., 2023) | 64 | 1388 | 0.01% |
| U-Tokyo xArm Pick and Place (Matsushima et al., 2023) | 92 | 6789 | 0.01% |
| Viola (Zhu et al., 2023b) | 135 | 68913 | 0.01% |
| Fractal (Brohan et al., 2022) | 87212 | 3786400 | 8.91 % |
| Furniture Bench Dataset (Heo et al., 2023) | 5100 | 3948057 | 0.51% |
| IAMLab CMU Pickup Insert (Saxena et al., 2023) | 631 | 146241 | 0.06% |
| Jaco Play (Dass et al., 2023) | 976 | 70127 | 0.10% |
| Kaist Non-prehensile (Kim et al., 2023) | 201 | 32429 | 0.02% |
| Kuka (Kalashnikov et al., 2018) | 209880 | 2455879 | 21.45% |
| NYU Door Opening Surprising Effectiveness (Pari et al., 2022) | 435 | 18196 | 0.04% |
| NYU ROT Dataset (Haldar et al., 2023) | 14 | 440 | 0.01% |
| RoboSet (Bharadhwaj et al., 2024) | 18250 | 1419999 | 1.86% |
| Roboturk (Mandlekar et al., 2019) | 1796 | 168423 | 0.18% |
| **RoboMIND (Wu et al., 2025a)** | **69274** | **21.4M** | **15%** |
| Single-Arm Franka | 16018 | 2268033 | 23.12% |
| Dual-Arm Franka | 1774 | 375807 | 2.56% |
| Single-Arm UR-5e | 25721 | 2643322 | 37.13% |

| Dataset | Episode | Frames | Weight |
|---|---|---|---|
| AgileX Cobot Magic V2.0 | 10059 | 6477564 | 14.52% |
| Tien Kung 1.0 | 15702 | 9683213 | 22.67% |
| **XR-D** | **158639** | **69.1M** | **35%** |
| Single-Arm Franka | 16933 | 5240845 | 10.67% |
| Dual-Arm Franka | 56800 | 17140497 | 35.80% |
| Single-Arm UR-5e | 21954 | 3218116 | 13.84% |
| Dual-Arm UR-5e | 33916 | 5463729 | 21.38% |
| AgileX Cobot Magic V2.0 | 8004 | 16576019 | 5.05% |
| ARX LIFT | 11866 | 15845836 | 7.48% |
| Tien Kung 1.0 | 9166 | 5605573 | 5.78% |
| **Ego4D (Grauman et al., 2022)** | **59427** | **14.3M** | **10%** |

## 6.4 ADDITIONAL REAL-WORLD EXPERIMENTS

Table 9: Success rate results across 20 tasks on Tien Kung 1.0.

| Method | TK1-Close Drawer | TK1-Flip TennisTube | TK1-Press CookerButton | TK1-Move ChopstickCup | TK1-Stack Cubes | TK1-Stack Cups | TK1-Stack Plates | TK1-Pick WipeTowel | TK1-Hang Towel | TK1-Open PotLid | - |
|---|---|---|---|---|---|---|---|---|---|---|---|
| UniVLA | 25 | 0 | 25 | 10 | 0 | 0 | 35 | 0 | 0 | 0 | - |
| RDT | 45 | 0 | 65 | 0 | 0 | 0 | 70 | 0 | 0 | 0 | - |
| GR00T-N1.5 | 75 | 20 | 85 | 20 | 0 | 0 | 70 | 0 | 0 | 0 | - |
| $\pi_0$ | 75 | 40 | 45 | 25 | 0 | 0 | 80 | 0 | 0 | 0 | - |
| $\pi_{0.5}$ | 75 | 35 | 55 | 60 | 0 | 0 | 75 | 0 | 0 | 0 | - |
| XR-1 (ours) | 80 | 50 | 90 | 65 | 65 | 20 | 85 | 55 | 65 | 20 | - |

| Method | TK1-Open Oven | TK1-Pack EggBox | TK1-Close Laptop | TK1-Insert Toaster | TK1-Flip Cup | TK1-Place FlipButton | TK1-Open TrashBin | TK1-Press Machine | TK1-Find Tape | TK1-Stack Bowls | Avg. |
|---|---|---|---|---|---|---|---|---|---|---|---|
| UniVLA | 30 | 0 | 0 | 30 | 0 | 0 | 25 | 20 | 20 | 30 | 12.5 |
| RDT | 0 | 0 | 90 | 0 | 0 | 0 | 85 | 55 | 0 | 0 | 20.5 |
| GR00T-N1.5 | 55 | 0 | 15 | 40 | 0 | 0 | 70 | 45 | 45 | 45 | 29.3 |
| $\pi_0$ | 75 | 10 | 90 | 65 | 10 | 15 | 75 | 70 | 70 | 80 | 41.3 |
| $\pi_{0.5}$ | 80 | 20 | 95 | 65 | 30 | 25 | 65 | 75 | 75 | 80 | 45.5 |
| XR-1 (ours) | 80 | 65 | 95 | 70 | 65 | 65 | 85 | 75 | 75 | 90 | 68.0 |

**Results on Tien Kung 1.0**. Table 9 reports success rates across 20 tasks on Tien Kung 1.0. XR-1 again outperforms all baselines by a clear margin. For example, in *TK1-HangTowel*, it achieves 65% success while all baselines fail (0%). Overall, XR-1 attains an average success rate of 68.0%, substantially higher than $\pi_0$ (41.3%) and more than double RDT (20.5%) and UniVLA (12.5%). These results highlight the effectiveness of UVMC supervision in providing robust representations and stable optimization across diverse manipulation skills.

Table 10: Success rate results across 20 tasks on Dual-Arm Franka.

| Method | DFR-Move CupMilk | DFR-Stack Bowls | DFR-Sweep Trash | DFR-Transfer Cup | DFR-Move Chopstick | DFR-Stack Cubes | DFR-Stack Plates | DFR-Clean Table | DFR-Hang CupHolder | DFR-Hang TowelRack | - |
|---|---|---|---|---|---|---|---|---|---|---|---|
| UniVLA | 15 | 20 | 0 | 30 | 0 | 5 | 25 | 35 | 0 | 20 | - |
| RDT | 55 | 40 | 0 | 0 | 15 | 60 | 90 | 65 | 0 | 0 | - |
| GR00T-N1.5 | 80 | 85 | 35 | 55 | 0 | 15 | 45 | 90 | 25 | 50 | - |
| $\pi_0$ | 0 | 75 | 35 | 60 | 0 | 75 | 80 | 80 | 0 | 25 | - |
| $\pi_{0.5}$ | 15 | 85 | 60 | 40 | 0 | 55 | 90 | 85 | 45 | 20 | - |
| XR-1 (ours) | 80 | 85 | 75 | 90 | 55 | 60 | 90 | 90 | 65 | 60 | - |

| Method | DFR-Find TapeBox | DFR-Pick ButtonPress | DFR-Sweep Rubbish | DFR-Close Toolbox | DFR-Collect BasketTea | DFR-Place Tools | DFR-Get Blocks | DFR-Place RagWipe | DFR-Open Toolbox | DFR-Place Screws | Avg. |
|---|---|---|---|---|---|---|---|---|---|---|---|
| UniVLA | 25 | 0 | 0 | 15 | 10 | 0 | 25 | 20 | 0 | 0 | 12.3 |
| RDT | 0 | 20 | 0 | 25 | 0 | 0 | 5 | 55 | 0 | 0 | 21.5 |
| GR00T-N1.5 | 80 | 0 | 0 | 85 | 35 | 0 | 70 | 0 | 0 | 0 | 37.5 |
| $\pi_0$ | 90 | 30 | 0 | 0 | 0 | 0 | 75 | 70 | 50 | 0 | 37.3 |
| $\pi_{0.5}$ | 65 | 25 | 30 | 0 | 0 | 60 | 75 | 70 | 0 | 0 | 41.0 |
| XR-1 (ours) | 90 | 75 | 60 | 90 | 75 | 60 | 85 | 70 | 60 | 55 | 73.5 |

**Results on Dual-Arm Franka**. Table 10 reports success rates across 20 tasks on the Dual-Arm Franka. XR-1 achieves the highest average performance (73.5%), substantially outperforming $\pi_0$ (37.3%) and other baselines. For example, in *DFR-TransferCup* it reaches 90% success, while all alternatives fall below 60%. It is because XR-1 leverages UVMC for richer supervision, yielding robust representations and stable learning across diverse objectives.

Table 11: Success rate results across 20 tasks on AgileX Cobot Magic V2.0.

| Method | AGX-Open DrawerButton | AGX-Move ButtonDrawer | AGX-Stack Boxes | AGX-Find TapeBox | AGX-Sweep Rubbish | AGX-Arrange Valves | AGX-Hang Scissors | AGX-Place Button | AGX-Close Toolbox | AGX-Gather Screws | - |
|---|---|---|---|---|---|---|---|---|---|---|---|
| UniVLA | 25 | 15 | 0 | 0 | 0 | 0 | 25 | 25 | 20 | 0 | - |
| RDT | 70 | 75 | 20 | 60 | 0 | 30 | 0 | 60 | 0 | 0 | - |
| GR00T-N1.5 | 85 | 75 | 20 | 75 | 0 | 45 | 0 | 80 | 0 | 0 | - |
| $\pi_0$ | 85 | 85 | 0 | 60 | 0 | 45 | 0 | 0 | 0 | 0 | - |
| $\pi_{0.5}$ | 35 | 70 | 40 | 80 | 15 | 35 | 0 | 70 | 35 | 30 | - |
| XR-1 (ours) | 90 | 80 | 45 | 75 | 25 | 45 | 80 | 85 | 90 | 30 | - |

| | AGX-Find Circuit | AGX-Place BiscuitBox | AGX-Collect BasketTea | AGX-Place Screwdriver | AGX-Pour GearOil | AGX-Stack BrakePads | AGX-Mesh StackCup | AGX-Pour Wine | AGX-Hang WipeRag | AGX-Stack Bowls | Avg. |
|---|---|---|---|---|---|---|---|---|---|---|---|
| UniVLA | 0 | 0 | 0 | 0 | 10 | 10 | 25 | 0 | 0 | 20 | 8.8 |
| RDT | 0 | 0 | 0 | 0 | 0 | 55 | 0 | 0 | 65 | 85 | 28.5 |
| GR00T-N1.5 | 0 | 50 | 45 | 0 | 0 | 0 | 0 | 0 | 20 | 70 | 24.0 |
| $\pi_0$ | 0 | 40 | 45 | 55 | 40 | 0 | 55 | 0 | 50 | 90 | 32.5 |
| $\pi_{0.5}$ | 20 | 60 | 25 | 30 | 55 | 40 | 15 | 20 | 60 | 90 | 41.3 |
| XR-1 (ours) | 15 | 60 | 35 | 35 | 75 | 85 | 90 | 20 | 55 | 85 | 60.0 |

**Results on AgileX Cobot Magic V2.0**. Table 11 reports success rates on 20 tasks with the AgileX Cobot Magic V2.0. XR-1 achieves an average of $60.0\%$, nearly doubling $\pi_0$ ($32.5\%$) and far surpassing UniVLA ($8.8\%$). On challenging tasks such as *AGX-StackBrakePads* and *AGX-CloseToolbox*, it reaches $85-90\%$, while other methods collapse to near $0\%$. We attribute these gains to UVMC-driven representations, which provide richer supervision and stabilize multi-task optimization.

Table 12: Success rate results across 20 tasks on Single-Arm UR-5e.

| Method | SUR-Find Tape | SUR-Move MilkCup | SUR-Stack Bowls | SUR-Open Drawer | SUR-Close Drawer | SUR-Insert ToyBlock | SUR-Place Chopstick | SUR-Stack Cubes | SUR-Stack Cup | SUR-Stack Plates | - |
|---|---|---|---|---|---|---|---|---|---|---|---|
| UniVLA | 35 | 25 | 50 | 20 | 35 | 0 | 0 | 0 | 0 | 0 | - |
| RDT | 80 | 35 | 20 | 35 | 45 | 0 | 0 | 0 | 0 | 0 | - |
| GR00T-N1.5 | 85 | 70 | 80 | 25 | 70 | 0 | 0 | 0 | 25 | 0 | - |
| $\pi_0$ | 25 | 55 | 90 | 50 | 85 | 0 | 0 | 80 | 0 | 55 | - |
| $\pi_{0.5}$ | 55 | 30 | 95 | 85 | 95 | 0 | 35 | 90 | 90 | 85 | - |
| XR-1 (ours) | 95 | 85 | 95 | 90 | 90 | 15 | 20 | 90 | 85 | 85 | - |

| | SUR-Slide Drawer | SUR-Open UpperDrawer | SUR-Open Oven | SUR-Pack EggBox | SUR-Close Laptop | SUR-Insert Bread | SUR-Assemble Valve | SUR-Pour TubeBeaker | SUR-Pour GearOil | SUR-Wipe HangRag | Avg. |
|---|---|---|---|---|---|---|---|---|---|---|---|
| UniVLA | 30 | 30 | 35 | 0 | 35 | 0 | 30 | 0 | 20 | 30 | 18.8 |
| RDT | 40 | 35 | 55 | 15 | 50 | 0 | 15 | 10 | 35 | 30 | 25.0 |
| GR00T-N1.5 | 45 | 65 | 80 | 0 | 90 | 0 | 80 | 0 | 10 | 30 | 37.8 |
| $\pi_0$ | 75 | 90 | 55 | 20 | 85 | 30 | 20 | 10 | 45 | 75 | 47.3 |
| $\pi_{0.5}$ | 90 | 90 | 95 | 80 | 90 | 85 | 25 | 10 | 50 | 75 | 67.5 |
| XR-1 (ours) | 80 | 90 | 90 | 70 | 90 | 65 | 90 | 20 | 85 | 75 | 75.3 |

**Results on Single-Arm UR-5e**. Table 12 summarizes success rates over 20 tasks on the Single-Arm UR-5e. XR-1 achieves the highest average success of $75.3\%$, clearly surpassing $\pi_0$ ($47.3\%$) and all other baselines. XR-1 maintains strong performance ($65\%$ and $85\%$) on hard tasks like *SUR-InsertBread* and *SUR-StackPlates* where baselines often collapse to near $0\%$. These results highlight the robustness and generalization ability of XR-1 enabled by UVMC.

Table 13: Success rate results across 20 tasks Dual-Arm UR-5e.

| Method | DUR-Find TapeBasket | DUR-Move CupMilk | DUR-Stack Bowls | DUR-Sweep Trash | DUR-Trans CupHolder | DUR-Stack Cubes | DUR-Hang CupHolder | DUR-Stack Brake | DUR-Sweep Rubbish | DUR-Press Button | - |
|---|---|---|---|---|---|---|---|---|---|---|---|
| UniVLA | 30 | 35 | 35 | 20 | 0 | 0 | 0 | 0 | 0 | 10 | - |
| RDT | 45 | 65 | 50 | 0 | 30 | 0 | 60 | 15 | 10 | 20 | - |
| GR00T-N1.5 | 25 | 60 | 80 | 0 | 0 | 0 | 0 | 0 | 0 | 0 | - |
| $\pi_0$ | 70 | 55 | 80 | 55 | 15 | 15 | 10 | 0 | 0 | 35 | - |
| $\pi_{0.5}$ | 45 | 65 | 90 | 60 | 55 | 20 | 10 | 85 | 70 | 80 | - |
| XR-1 (ours) | 85 | 65 | 65 | 60 | 20 | 85 | 65 | 80 | 85 | | - |

| | DUR-Pick PlaceTape | DUR-Close Toolbox | DUR-Assemble Valve | DUR-Flip TennisTube | DUR-Place Tools | DUR-Close DoorKnob | DUR-Take BasketTea | DUR-Pick Toolbox | DUR-Open TrashBin | DUR-Trans Buttons | Avg. |
|---|---|---|---|---|---|---|---|---|---|---|---|
| UniVLA | 35 | 30 | 25 | 20 | 30 | 40 | 25 | 30 | 30 | 0 | 19.8 |
| RDT | 85 | 10 | 0 | 35 | 0 | 20 | 85 | 0 | 0 | 0 | 26.5 |
| GR00T-N1.5 | 55 | 0 | 50 | 45 | 65 | 80 | 90 | 55 | 85 | 0 | 34.5 |
| $\pi_0$ | 20 | 85 | 0 | 70 | 35 | 85 | 80 | 70 | 55 | 20 | 42.8 |
| $\pi_{0.5}$ | 65 | 90 | 45 | 70 | 65 | 50 | 90 | 75 | 75 | 35 | 62.0 |
| XR-1 (ours) | 85 | 90 | 55 | 80 | 65 | 90 | 90 | 75 | 85 | 30 | 72.0 |

**Results on Dual-Arm UR-5e**. In addition to the bar plot reported in Figure 4, we provide the corresponding numerical results in Table 13. The table summarizes success rates across 20 tasks on the Dual-Arm UR-5e, offering a more detailed comparison among different methods.

## 6.5    ADDITIONAL GENERALIZATION ANALYSIS

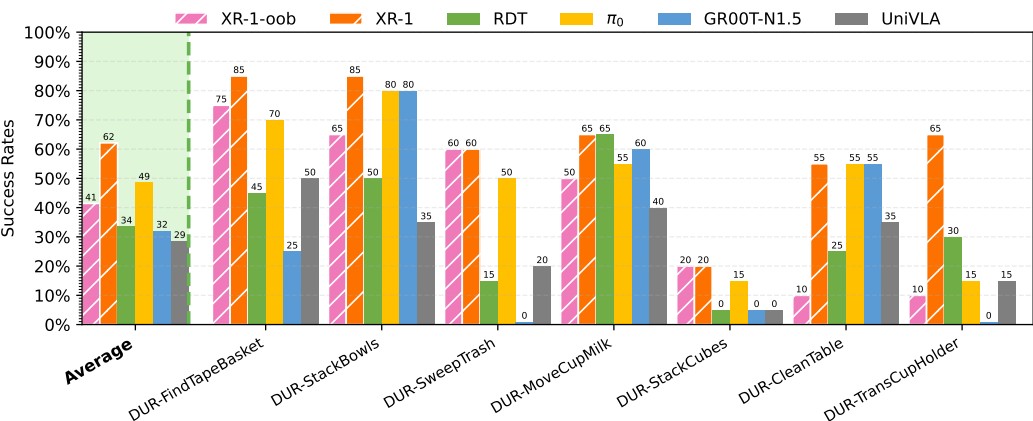

Figure 9: Out-of-box evaluation results of 7 tasks on Dual-Arm UR-5e.

**Out-of-Box Evaluation**. In addition to the evaluation on the Dual-Arm UR-5e, we also conduct an out-of-box evaluation of XR-1 on the Dual-Arm Franka. Specifically, we select 7 representative tasks from XR-D, covering only $0.9\%$ of the dataset. To ensure a fair comparison, baselines without XR-D pretraining are fine-tuned on data from these tasks prior to evaluation. As shown in Figure 9, the pretrained **XR-1-oob** model, even without Stage-3 task-specific adaptation, achieves performance comparable to $\pi_0$, while consistently outperforming GR00T-N1.5, RDT, and UniVLA. This result highlights XR-1's strong generalization ability in low-data regimes.

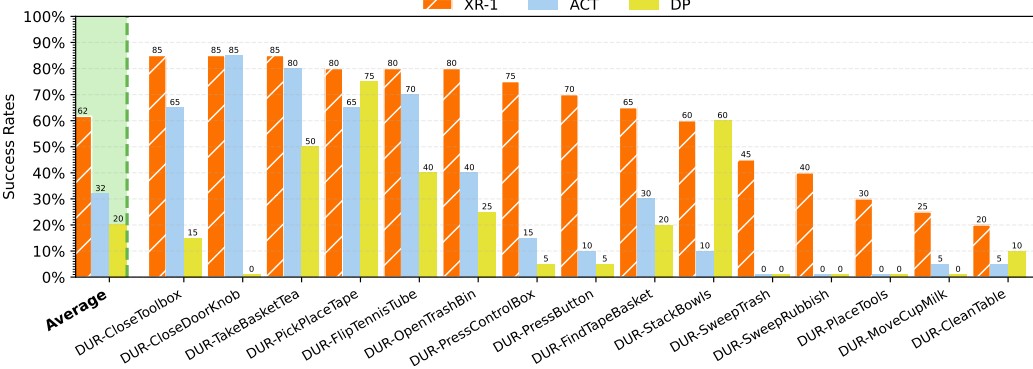

Figure 10: Fast adaption on Dual-Arm UR5e. Dual-Arm UR5e is an embodiment included in XR-D. In this setup, Here, XR-1 adapts to 15 novel tasks with one model using only 20-shot demonstrations per task, while baselines (ACT and DP) are trained per task.

**Fast Adaptation to New Tasks**. Beyond the experiments on Tien Kung 2.0, we also evaluate fast adaptation on the Dual-Arm UR-5e. Specifically, we collect 15 new tasks that are unseen in XR-D, each with 20 trajectories for training. XR-1 is trained jointly across these tasks, while single-task baselines, ACT (Zhao et al., 2023) and Diffusion Policy (DP) (Chi et al., 2023), are trained independently per task. As shown in Figure 10, XR-1 achieves substantially higher success rates than ACT and DP, even though the evaluation setting is more favorable to the baselines. This performance gain can be attributed to large-scale pretraining combined with UVMC supervision, which enables XR-1 to extract transferable representations from few-shot data and adapt effectively across diverse manipulation tasks.

### 6.6    ADDITIONAL ABLATION STUDY

**Ablation study of UVMC.** To obtain deeper insights into the UVMC architecture and its key hyperparameter choices, we conduct 10 ablation experiments, as summarized in Table 14, with Exp.10

serving as the baseline. By comparing Exp.1–6 against Exp.10, we analyze the influence of different codebook category numbers and code dimensions on the final performance. For the code dimension, Exp.1–3 adopt 64, 128, and 512, respectively, and are compared with the baseline setting of 256 in Exp.10. The results show that when the dimension is below 256, policy performance consistently improves as the dimension increases from 64 to 128 and 256, while further increasing it to 512 yields no clear additional gains, suggesting that a dimension of 256 is already near-optimal. Using a similar protocol in Exp.4–6 for the category number, we finally adopt 256 categories and a 256-dimensional codebook as our default configuration. Next, by comparing Exp.7–8 with Exp.10, we evaluate the difference between using only motion codes, only vision codes, and the unified vision–motion code (UVMC). The results indicate that both motion-only and vision-only variants underperform the unified UVMC. Moreover, vision-only codes outperform motion-only codes, while combining both modalities within UVMC leads to complementary effects and improved overall performance. Finally, by comparing Exp.9 and Exp.10, we investigate whether a combined or separate codebook is more effective. The results show that both designs achieve comparable performance, which we attribute to the alignment loss imposed during training: although a separate codebook increases the number of learnable codes, the alignment constraint effectively regulates cross-modal relationships, leading to similar execution capabilities for both schemes.

Table 14: Ablation study of UVMC.

| Exp. | Codebook | Category×Embed.Dim | UVMC Token | Stage-1&2&3 | DUR-Clean Table | DUR-Find TapeBasket | DUR-Move CupMilk | DUR-Stack Bowls | DUR-Sweep Trash | DUR-Trans CupHolder | Avg |
|---|---|---|---|---|---|---|---|---|---|---|---|
| 1 | combine | 256×64 | both | DT | 35 | 55 | 50 | 60 | 35 | 45 | 46.7 |
| 2 | combine | 256×128 | both | DT | 45 | 65 | 60 | 70 | 55 | 65 | 60.0 |
| 3 | combine | 256×512 | both | DT | 55 | 75 | 50 | 85 | 65 | 55 | 64.2 |
| 4 | combine | 64×256 | both | DT | 45 | 60 | 55 | 65 | 35 | 50 | 51.7 |
| 5 | combine | 128×256 | both | DT | 50 | 65 | 55 | 65 | 60 | 60 | 59.2 |
| 6 | combine | 512×256 | both | DT | 40 | 80 | 60 | 80 | 55 | 65 | 63.3 |
| 7 | combine | 256×256 | motion-only | DT | 25 | 70 | 35 | 60 | 5 | 15 | 35.0 |
| 8 | combine | 256×256 | vision-only | DT | 10 | 70 | 65 | 70 | 15 | 65 | 50.0 |
| 9 | separate | 256×256 | both | DT | 55 | 75 | 55 | 85 | 40 | 80 | 65.0 |
| 10 | combine | 256×256 | both | DT | 50 | 75 | 65 | 80 | 60 | 70 | 66.7 |

**Ablation study of Ego4d.** To further examine the contribution of human video data (Ego4D) in the pre-training stage, we conduct a set of ablation experiments, as summarized in Table 15. To balance computational cost and the reliability of the conclusions, we use 10% of the full pre-training dataset for these comparisons. Under this setting, we evaluate two variants: one with Ego4D included in the pre-training data and one without Ego4D (w/o Ego4D). As shown in Table 15, removing Ego4D leads to a 5.8% drop in average success rate compared to the setting that includes Ego4D. These results quantitatively suggest that incorporating Ego4D into the pre-training data can effectively improve performance.

Table 15: Ablation study of Ego4d.

| Exp. | Instantiation | Stage-1 | Stage-2 | Stage-3 | DUR-Clean Table | DUR-Find TapeBasket | DUR-Move CupMilk | DUR-Stack Bowls | DUR-Sweep Trash | DUR-Trans CupHolder | Avg |
|---|---|---|---|---|---|---|---|---|---|---|---|
| 1 | XR-1 w/o Ego4D | 10% | XR-D | ✓ | 20 | 60 | 10 | 55 | 15 | 35 | 32.5 |
| 2 | XR-1 w/ Ego4D | 10% | XR-D | ✓ | 25 | 60 | 25 | 60 | 20 | 40 | 38.3 |

**Cross-Embodied Knowledge Transfer for Enhanced Single Embodiment Performance**. This setup is designed to verify whether similar tasks across different embodiments can mutually benefit each other. Since the UVMC counterpart of XR-1 learns an embodiment-agnostic feature, this setup serves to validate that capability. Specifically, we selected two identical tasks (FindTape and SweepRubbish) across three different embodiments (Dual-Arm Franka, Dual-Arm UR5e, and Tien Kung 2.0). The detailed results are shown in Table 16. Exp. 2 represents the results of training these two skills across three different embodiments, resulting in six tasks. In the comparative experiment setup, training two skills for a specific embodiment typically results in only two tasks. Therefore, to ensure fairness, in Exp. 1, we added four additional tasks for the same embodiment, ensuring that the data volume is equivalent. *The final results indicate that learning the same skills across different embodiments can enhance the success rate of each embodiment's skills, increasing the average success rate by approximately 15%.* This demonstrates that the UVMC module has learned an embodiment-agnostic beneficial feature.

Table 16: Ablation study of XR-1 on cross-embodiment knowledge transfer.

| Exp. | Instantiation | Stage-1 | Stage-2 | Stage-3 | DFR-Find TapeBox | DFR-Sweep Rubbish | DUR-Pick PlaceTape | DUR-Sweep Rubbish | TK2-Take Tape | TK2-Sweep Rubbish | Avg. |
|---|---|---|---|---|---|---|---|---|---|---|---|
| 1 | XR-1 | 100% | XR-D | SelfRobot | 50 | 20 | 70 | 50 | 60 | 30 | 47 |
| 2 | XR-1 | 100% | XR-D | CrossRobot | 70 | 30 | 70 | 60 | 70 | 70 | 62 |

## 6.7 VISUALIZATION OF UNIFIED VISION-MOTION CODES

### 6.7.1 FRAME-TO-FRAME NEAREST-NEIGHBOR RETRIEVAL BETWEEN MOTION AND VISION CODES

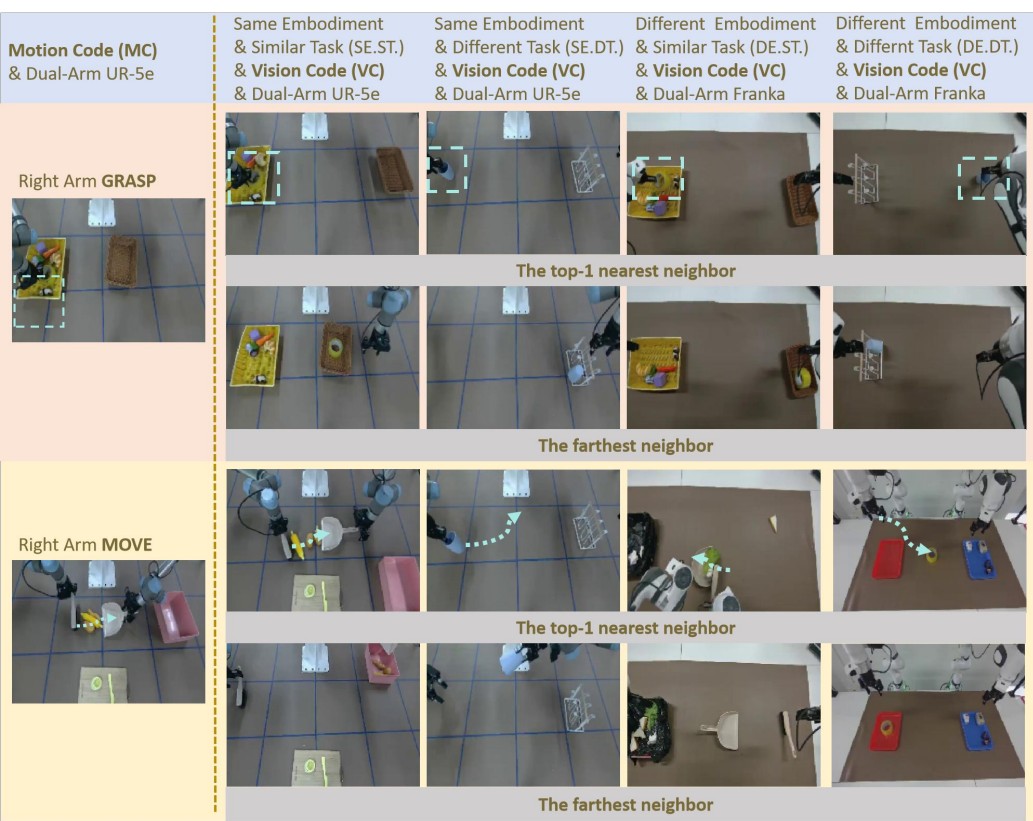

Figure 11: **Frame-to-Frame Nearest-neighbor Retrieval between motion and vision codes.** In the first column, we select two representative skill actions, grasp and move, and compute their Motion Codes (MC). Columns two through five report the cosine distances between Motion Code (MC) and Vision Code (VC) features and their nearest and farthest neighbors under four settings: Same Embodiment with a Similar Task (SE.ST.), Same Embodiment with a Different Task (SE.DT.), Different Embodiment with a Similar Task (DE.ST.), and Different Embodiment with a Different Task (DE.DT.). For the different-embodiment setting, we use a dual-arm Franka robot. To standardize the notion of left and right across embodiments, we define them with respect to the outward-facing direction of the embodiment.

To further evaluate whether the vision code (VC) and motion code (MC) are semantically aligned in the latent space, we design a nearest-neighbor retrieval experiment, as shown in Figure 11. We select 7 tasks, each with 10 trajectory episodes, and compute the VC and MC for all frames in each episode. At the current time step $T$, we extract the MCs corresponding to the actions GRASP and MOVE, as illustrated in the first column of Figure 11, and visualize the image of the current frame to better interpret the motion code semantics. We then compute VC features from different tasks and embodiments and, using cosine similarity, identify for each MC feature at time $T$ its nearest-neighbor and farthest-neighbor VC feature. Columns two through five of Figure 11 report

the cosine distances between MC and VC features under four settings: Same Embodiment with a Similar Task (SE.ST.), Same Embodiment with a Different Task (SE.DT.), Different Embodiment with a Similar Task (DE.ST.), and Different Embodiment with a Different Task (DE.DT.). For the different-embodiment setting, we use a dual-arm Franka robot. To standardize the notion of left and right across embodiments, we define them with respect to the outward-facing direction of the embodiment.

Comparing SE.ST. with GRASP and MOVE shows that, under the same embodiment and a similar task, the nearest-neighbor VC for a given MC consistently reflects the same action semantics, whereas the farthest-neighbor images display clearly different motions. Under the SE.DT. setting, despite task changes, the nearest-neighbor VCs still capture the semantics of grasp or move, while the farthest-neighbor images correspond to distinct motions. In the DE.ST. setting, even with different embodiments, the nearest-neighbor frames for a similar task consistently depict similar actions, in contrast to the clearly different motions in the farthest-neighbor images. Likewise, in the DE.DT. setting, nearest-neighbor retrieval continues to select frames whose semantics are closest to grasp or move, despite differences in both embodiment and task, whereas the farthest-neighbor images represent dissimilar actions. Together, these visualizations demonstrate that UVMC successfully learns semantic representations of different actions, and that these representations become embodiment-agnostic.

### 6.7.2 TASK-TO-TASK NEAREST-NEIGHBOR RETRIEVAL BETWEEN MOTION AND VISION CODES

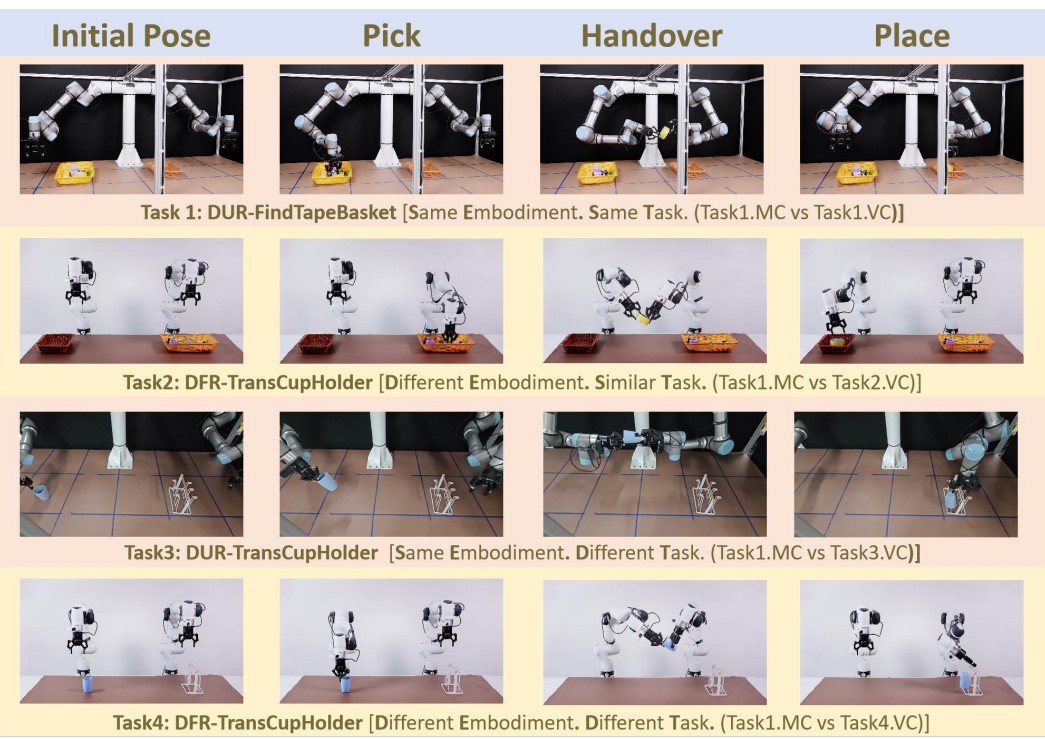

Figure 12: **Task-to-task nearest-neighbor retrieval between motion and vision codes.** It illustrates four different tasks and annotates the corresponding process phase at each timestep. For each task, the annotations correspond one-to-one to the four experimental settings in Table 17, representing the actual procedure used to compute nearest-neighbor retrieval across different tasks.

In the preceding analysis (Appendix 6.7.1), we primarily evaluate nearest-neighbor retrieval performance at the frame-to-frame level for representative skills (e.g., grasp, move) under different experimental settings (SE.ST., SE.DT., DE.ST., DE.DT.). To further investigate cross-task similarity from an episode-to-episode perspective, we construct the nearest-neighbor similarity distribution as illustrated in Figure 12. Specifically, we select four distinct tasks and perform pairwise com-

Table 17: Task-to-task Nearest-neighbor similarity statistics

| Exp. | The mode of similarity distribution | Average |
|---|---|---|
| SE.ST. (Task1.MC vs Task1.VC) | 0.97 | 0.75 |
| DE.ST. (Task1.MC vs Task2.VC) | 0.82 | 0.67 |
| SE.DT. (Task1.MC vs Task3.VC) | 0.60 | 0.56 |
| DE.DT. (Task1.MC vs Task4.VC) | 0.46 | 0.48 |

parisons among them, following the same experimental configurations as before (SE.ST., SE.DT., DE.ST., DE.DT.). Specifically, let the motion code of the $i$-th frame in the source task be denoted by $MC_i$, and the vision code of the $j$-th frame in the target task be denoted by $VC_j$. We first compute the similarity between each $MC_i$ in the source task and all $VC_j$ in the target task:

$$\mathbf{S} \in \mathbb{R}^{T_s \times T_t}, \quad \mathbf{S}_{i,j} = \mathrm{sim}(MC_i, VC_j),$$

where $T_s$ and $T_t$ denote the number of frames in the source and target tasks, respectively. Based on this matrix $\mathbf{S}$, we perform a column-wise maximization, i.e., for each target frame $j$, we select from all source frames the motion-vision pair that attains the highest similarity:

$$s_j = \max_{1 \le i \le T_s} \mathbf{S}_{i,j}.$$

In this way, we reduce the original two-dimensional similarity matrix to a similarity vector:

$$\mathbf{s} = [s_1, s_2, \ldots, s_{T_t}],$$

which characterizes, for each target frame, its nearest-neighbor similarity with the source task. Finally, we normalize the elements of $\mathbf{s}$ to obtain a normalized nearest-neighbor similarity vector $\tilde{\mathbf{s}}$, whose values are constrained to lie within $[0, 1]$, enabling comparable and stable statistical analysis across different tasks.

In Table 17, we report pairwise comparison results across different tasks. We take task 1 as the reference and use cosine similarity to quantify representational similarity between tasks. Under the SE.ST. setting, the mode of the similarity distribution is close to 1, indicating that for identical embodiments and tasks the model learns highly consistent representations between MC and VC. Among the remaining three settings, DE.ST. has the highest mode, suggesting that UVMC learns features that are largely independent of embodiment and instead capture action-centric skills. Comparing DE.ST. with SE.DT. further supports this: different-embodiment but similar-task pairs exhibit higher overall similarity than same-embodiment but different-task pairs, implying stronger semantic alignment for shared skills than for shared morphology alone. Although DE.DT. has the lowest mode, it still retains non-trivial similarity, indicating that shared low-level skills (such as pick, handover, and place) give rise to stable cross-task, cross-embodiment similarity in the learned representations. We also compute the average nearest-neighbor similarity under these settings and observe consistent conclusions. Consequently, these analyses demonstrate that UVMC learns semantically meaningful action representations that are largely invariant to embodiment.

### 6.7.3 VISUALIZING UVMC WITH T-SNE

To qualitatively validate whether UVMC effectively captures intrinsic task dynamics and abstracts away physical embodiment details, we employ t-SNE to project the high-dimensional latent embeddings into a two-dimensional manifold. We conduct this analysis on two distinct subsets: (1) a single-robot scenario involving 6 tasks performed by a dual-arm UR robot (Figure 13), and (2) a mixed-embodiment scenario comprising both dual-arm Franka and dual-arm UR robots (Figure 14). The visualization reveals two critical properties of the representation learned by UVMC.

**Semantic Consistency of Dynamics.** As shown in Figure 13, the embedding space exhibits a structured organization where tasks characterized by similar motion primitives form cohesive, proximal clusters. Specifically, in the lower-right corner, tasks sharing the "left-arm pick" primitive—namely DUR-CleanTable, DUR-StackBowls, and DUR-SweepTrash—are grouped closely together. This suggests the model successfully encodes the shared underlying semantics of the grasping motion. Conversely, the model effectively isolates distinct behaviors. Observing the upper region of the plot,

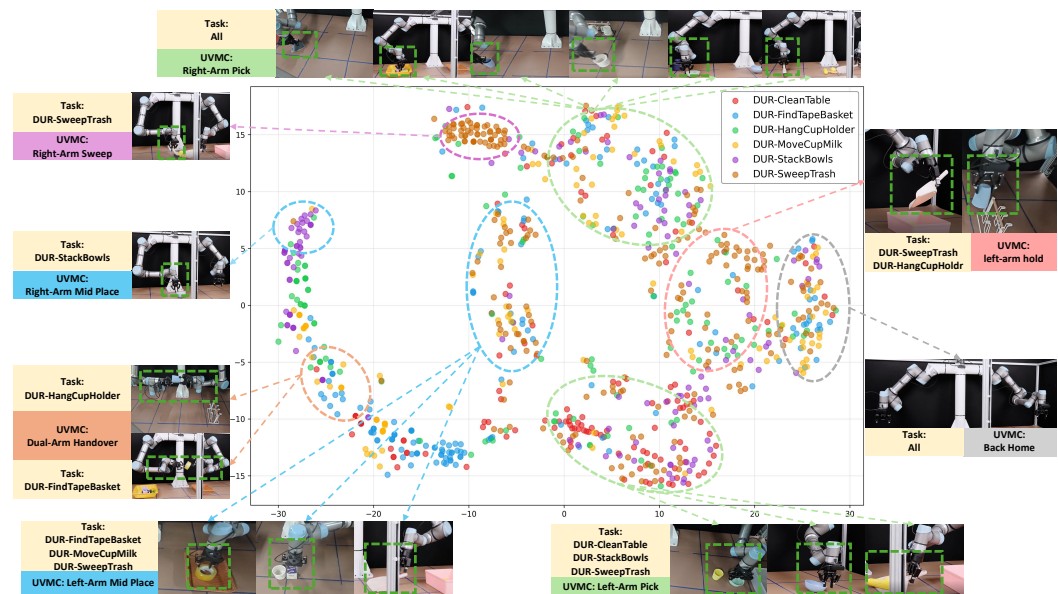

Figure 13: Visualizing UVMC on 6 dual-arm UR tasks using t-SNE.

a cluster of brown points forms a distinct island separate from other task embeddings. This cluster corresponds to the sweeping and translational motions unique to the DUR-SweepTrash task. This separation demonstrates that UVMC can effectively disentangle common dynamical patterns (e.g., picking) from task-specific nuances (e.g., sweeping).

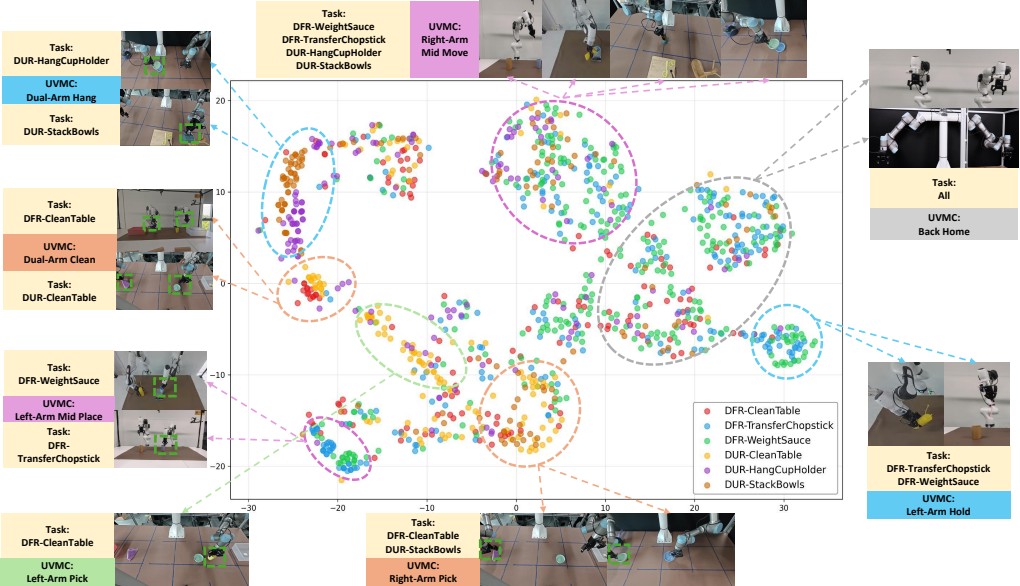

Figure 14: Visualizing UVMC across different embodiments (Dual-Arm Franka and Dual-Arm UR) using t-SNE.

**Cross-Embodiment Alignment.** A central hypothesis of our approach is that the learned representation should be embodiment-agnostic. Figure 14 substantiates this by illustrating the embedding space for identical tasks executed by morphologically distinct robots. Notably, the embeddings for DFR-CleanTable (represented in red) and DUR-CleanTable (represented in yellow) share a common support and overlap significantly within the latent manifold. Despite the kinematic and appearance discrepancies between the Franka and UR robots, UVMC projects their state-action trajectories onto

a unified manifold. This result indicates that the model has learned a robust, embodiment-invariant representation that prioritizes high-level task semantics over low-level proprioceptive differences.

### 6.8 FAILURE CASE ANALYSIS ON BASELINE METHODS

We conducted a qualitative analysis of the rollout videos to investigate why baselines struggle compared to XR-1. We categorize the observed failures into two primary modes:

**Optimization Collapse and Conflicting Gradients**. In some distinct scenarios, baselines fail to capture the correct motion trend entirely. The models exhibit "hesitation" or revert to the mean pose, suggesting that the optimization objective is torn between conflicting gradients from different tasks. Example: In the 'DFR-CloseToolbox' task, the $\pi_0$ policy initiates a downward movement with the right arm but immediately retracts to the initial position. The robot appears indecisive and fails to commit to the task trajectory. We attribute this to the difficulty of fitting a single policy distribution to 20 diverse tasks without task-distinguishing representations.

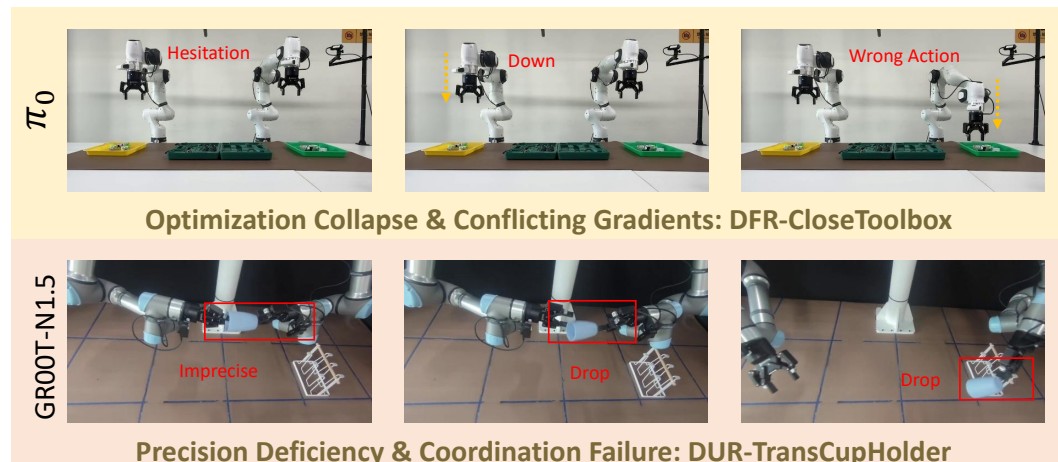

Figure 15: Failure cases of baseline methods.

**Precision Deficiency and Coordination Failure**. The most common failure mode involves the robot attempting the correct action but failing in execution precision or bimanual coordination. Example: In the 'DUR-HangCupHolder' task, GR00T-N1.5 successfully grasps the cup with the right arm. However, it drops the cup during the handover to the left arm. This indicates that while the model learns the general policy distribution, it lacks the fine-grained control and temporal consistency required for complex, multi-stage manipulation.

XR-1 addresses these issues through the Unified Vision-Motion Condition (UVMC) introduced in Stage 1. The UVMC serves as a compact representation of visual dynamics and motion patterns. By conditioning the policy on UVMC, XR-1 can explicitly distinguish between different task modes, thereby reducing gradient conflicts during multi-task optimization. As an intermediate feature supervision signal, UVMC guides the model to generate smoother and more physically consistent actions. This additional supervision is critical for tasks requiring high precision (e.g., dual-arm handover), preventing the coordination failures observed in baselines like GR00T.

### 6.9 FAILURE CASE ANALYSIS ON XR-1

**Precision Deficiency**. The most common failure mode involves the robot attempting the correct action but failing in execution precision or bimanual coordination. Example: In the 'TK2-CollectScrews' task, the robot may fail to grasp a tiny screw or drop it mid-motion due to slight localization errors. This reflects the inherent difficulty of learning precise, bimanual coordination for dexterous manipulation tasks.

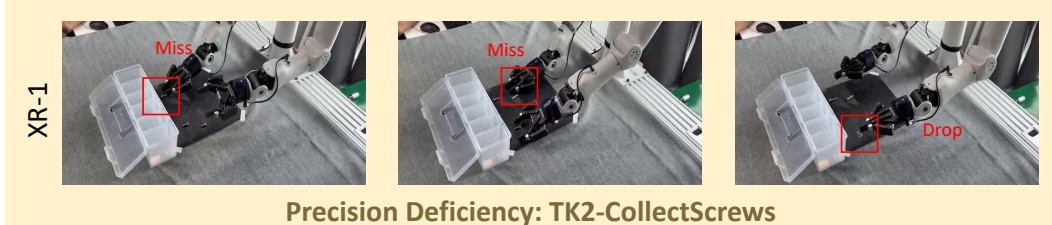

Figure 16: Failure Cases of XR-1.

## 6.10 SIMULATION BENCHMARK

We conduct simulation experiments on SimplerEnv (Li et al., 2024), a real-to-simulation manipulation benchmark. Concretely, policies are trained on real-robot data from BridgeData V2 (Walke et al., 2023) and then evaluated in SimplerEnv, as illustrated in the left part of Figure 17.To evaluate the effectiveness of our approach, especially UVMC, we compare XR-1 against the baseline $\pi_0$. In these experiments, we set the action chunk size to 4 and train for 60k iterations while keeping all other hyperparameters identical to those used for real-robot experiments. As shown in the right part of Figure 17, XR-1 achieves a higher average success rate than $\pi_0$, with an overall improvement of 27%. This result demonstrates that our method provides consistent performance gains even in the simulation setting.

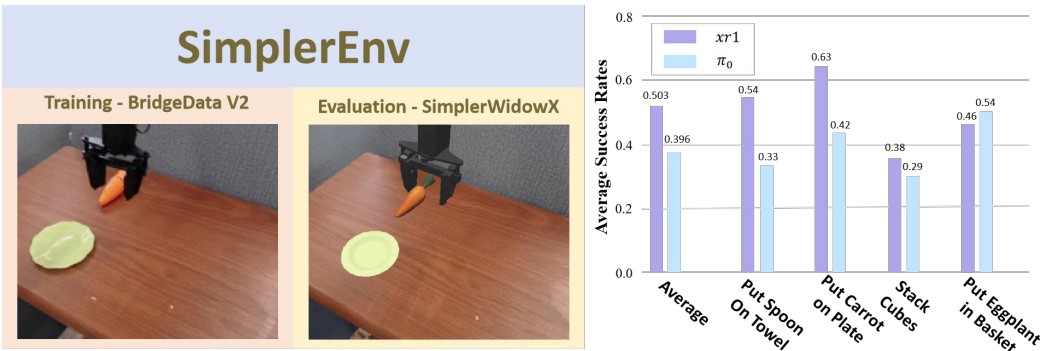

Figure 17: **Simulation benchmark – SimplerEnv.** The left panel illustrates one SimplerEnv setting, where policies are pretrained on real-robot data from BridgeData V2 and then evaluated in the simulation environment. The right panel reports the test performance of XR-1 and $\pi_0$ on SimplerEnv.

## 6.11 REPRESENTATIVE TASKS

As illustrated in Figure 18, we select a set of representative tasks from real-world experiments to provide detailed descriptions of the evaluation scenarios. These tasks are designed to cover a broad spectrum of challenges, including bimanual collaboration, dexterous manipulation, fluid/deformable object handling, contact-rich interactions, dynamic environments, and long-horizon manipulation. Together, they demonstrate the versatility and robustness of **XR-1** across diverse manipulation settings.

- **Bimanual Collaboration:** *DUR-TransCupHolder*. This task involves a coordinated bimanual operation: the right arm initially grasps a cup, performs an aerial handover to the left arm, which subsequently places the cup into a cup rack.
- **Dexterous Manipulation:** *DUR-CloseDoorKnob*. The robot performs a dexterous operation to close and lock the control box door. The right arm first manipulates the door to a closed position. Subsequently, the left arm rotates the door handle by 90 degrees and presses it inward to engage the locking mechanism.

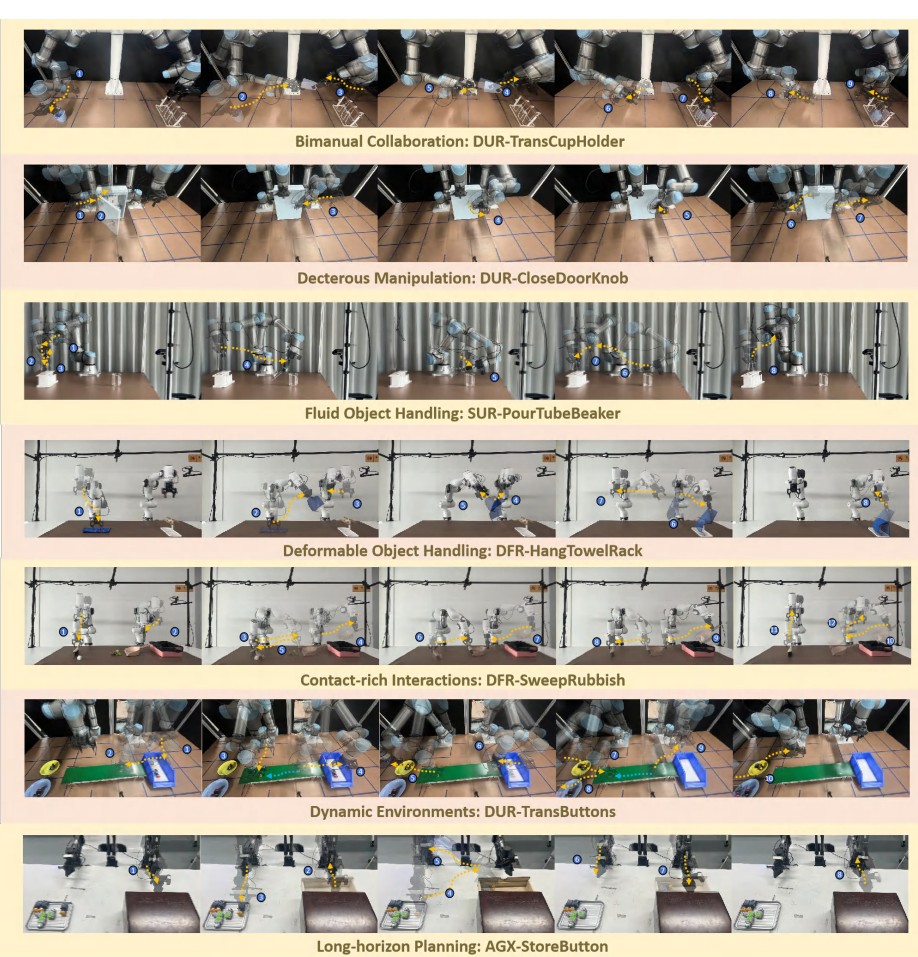

Figure 18: Diverse task settings in evaluation: bimanual collaboration, dexterous manipulation, deformable object handling, contact-rich interactions, dynamic environments, and long-horizon tasks.

- **Fluid Object Handling:** *SUR-PourTubeBeaker*. The task consists of three phases: removing a test tube from the rack, pouring its liquid into a measuring cup, and returning the test tube to the rack.

- **Deformable Object Handling:** *DFR-HangTowelRack*. The robot performs a bimanual manipulation task involving deformable object handling: the right arm first picks up a towel from a surface and transfers it to the left arm via an aerial handover; the left arm then manipulates the towel to drape it over a towel rack, completing the hanging motion.

- **Contact-Rich Interactions:** *DFR-SweepRubbish*. A dual-arm cleaning task is executed where the right arm operates a broom and the left arm stabilizes a dustpan. The robot systematically sweeps food remnants and a crumpled paper ball into the dustpan, followed by transporting and emptying the dustpan into a waste bin after each collection.

- **Dynamic Environments:** *DUR-TransButtons*. The robot's left arm loads colored button workpieces onto a moving conveyor belt, while the right arm autonomously identifies each part's color upon arrival and places it into the respective color-matched container.

- **Long-Horizon Manipulation:** *AGX-StoreButton*. This task entails a sequential dual-arm interaction: the left arm opens a drawer and holds it open, enabling the right arm to place a button workpiece inside; the left arm then closes the drawer after object deposition.

### 6.11.1 DATASET FOR EVALUATION

The dataset is primarily employed for the final fine-tuning stage of XR-1, and for training and evaluation of multiple baselines on this benchmark.

Table 18: The tasks summary of our real-world experiments.

| # | Task | Trajectory Num. | Avg. Len. | Task Instruction | Task Setting |
|---|---|---|---|---|---|
| | | **Dual-Arm UR5e** | | | |
| T1 | DUR-FindTapeBasket | 160 | 155 | Find the packaging tape and put it into the other basket |  |
| T2 | DUR-MoveCupMilk | 198 | 149 | Place the cup in the middle of the table and pick up the milk and place it next to the cup. |  |
| T3 | DUR-StackBowls | 158 | 147 | Put the blue bowl in the middle of the table and stack the green bowl on top of it |  |
| T4 | DUR-SweepTrash | 192 | 293 | Sweep up the rubbish and take out the trash |  |
| T5 | DUR-TransCupHolder | 167 | 170 | Pick up the cup with the right arm, hand it over to the left arm, and hang it on the holder with the left arm |  |
| T6 | DUR-StackCubes | 158 | 153 | Put the blue cube in the middle of the desk and stack it on top of the other blue cube |  |

| # | Task | Trajectory | | Task Instruction | Task Setting |
|---|------|------|------|------|------|
| | | Num. | Avg. Len. | | |
| T7 | DUR-HangCupHolder | 167 | 223 | Hang the cup on the holder |  |
| T8 | DUR-StackBrake | 200 | 81 | Use the left arm to place Brake Pad Type A in the middle, then use the right arm to pick up Brake Pad Type B and stack it on top of Brake Pad Type A |  |
| T9 | DUR-SweepRubbish | 185 | 157 | Sweep up the rubbish |  |
| T10 | DUR-PressButton | 183 | 121 | Pick up and place the green button, then press it |  |
| T11 | DUR-PickPlaceTape | 192 | 111 | Pick up and place the adhesive tape |  |
| T12 | DUR-CloseToolbox | 114 | 190 | Use both arms to close the toolbox |  |
| T13 | DUR-AssembleValve | 253 | 108 | Assemble the valve |  |
| T14 | DUR-FlipTennisTube | 123 | 127 | Put the tennis tube upright |  |
| T15 | DUR-PlaceTools | 198 | 102 | Use the left arm to place the screwdriver on the left side of the toolbox, and use the right arm to place the wrench on the right side of the toolbox |  |
| T16 | DUR-CloseDoorKnob | 201 | 147 | The right arm closes the distribution box door, and the left arm turns and presses the closing knob |  |
| T17 | DUR-TakeBasketTea | 198 | 147 | The right arm places the shopping basket in the middle, while the left arm takes tea drinks from the shelf and puts them inside |  |
| T18 | DUR-PickToolbox | 200 | 117 | Use both arms to pick up the toolbox |  |

| # | Task | Trajectory | | Task Instruction | Task Setting |
|---|------|------|------|---|---|
| | | Num. | Avg. Len. | | |
| T19 | DUR-OpenTrashBin | 28 | 45 | Open the trash bin |  |
| T20 | DUR-TransButtons | 146 | 184 | Transport the buttons from left to right and place them in the corresponding plates |  |
| T21 | DUR-CleanTable | 197 | 183 | Move the buttons from left to right and place them on the corresponding plates |  |
| | | | **Tien Kung 2.0** | | |
| T1 | TK2-PressButton | 291 | 292 | The left arm picks up the green button and places it in the middle, while the right arm presses it |  |
| T2 | TK2-AssembleValve | 162 | 382 | Assemble the valve |  |
| T3 | TK2-StackBrake | 178 | 261 | Use the left arm to place Brake Pad Type A in the middle, and use the right arm to pick up Brake Pad Type B and stack it on top of Brake Pad Type A |  |
| T4 | TK2-PlaceCircuit | 209 | 359 | The left arm picks up the circuit breaker from the red tray and places it in the middle of the table. Then the right arm picks up the circuit breaker and puts it into the blue tray on the right. |  |
| T5 | TK2-PressControlBox | 256 | 292 | The left arm places the control box in the middle, while the right arm presses the red emergency stop button on it |  |
| T6 | TK2-CloseDoorKnob | 197 | 271 | The right arm closes the door of the distribution box, and the left arm rotates and presses the closing knob. |  |
| T7 | TK2-GatherTools | 177 | 452 | Use the left arm to place the screwdriver on the left side of the toolbox, and use the right arm to place the wrench on the right side |  |
| T8 | TK2-CloseLaptop | 189 | 186 | Close the laptop |  |
| T9 | TK2-OpenPotLid | 190 | 304 | Open the blue pot lid |  |

| # | Task | Trajectory | | Task Instruction | Task Setting |
|---|---|---|---|---|---|
| | | Num. | Avg. Len. | | |
| T10 | TK2-HangCupHolder | 144 | 277 | Hang the oval-bottom cup on the holder |  |
| T11 | TK2-MoveCupSauce | 129 | 312 | Move the blue cup, pick up the yellow sauce bottle, and pour it into the blue cup |  |
| T12 | TK2-StackCup | 161 | 291 | Stack the blue cups |  |
| T13 | TK2-InsertToyBlock | 152 | 521 | Insert the blue toy into the square-bottom slot of the grey block |  |
| T14 | TK2-FindCapacitor | 232 | 370 | The left arm picks up the red electrolytic capacitor from the blue tray and places it in the middle of the table. Then the right arm picks it up and puts it into the red tray on the right |  |
| T15 | TK2-MoveMilkMug | 279 | 284 | Pick up and place the milk, then move the white mug |  |
| T16 | TK2-PourGearOil | 152 | 503 | The left arm places the gear on the middle metal tray, while the right arm pours lubricating oil on it |  |
| T17 | TK2-PlaceBiscuitBox | 133 | 447 | Pick up the biscuit box from the blue basket with the right arm and place it in the middle of the table. Then, use the left arm to place it on the middle shelf of the black rack |  |
| T18 | TK2-CollectScrews | 192 | 620 | The right arm places the two long screws into the slot at the very right end of the storage box, while the left arm places the two short screws into the slot at the very left end of the storage box |  |
| T19 | TK2-MoveTape | 182 | 547 | Pick up and place the rattan basket |  |
| T20 | TK2-TakeBasketTea | 197 | 497 | The right arm places the shopping basket in the middle, while the left arm takes tea drinks from the shelf and puts them inside |  |

| # | Task | Trajectory | | Task Instruction | Task Setting |
| | | Num. | Avg. Len. | | |
|---|---|---|---|---|---|
| T21 | TK2-TakeTape | 297 | 357 | Pick up and place the adhesive tape | |
| T22 | TK2-SweepRubbish | 239 | 421 | Sweep up the rubbish | |
| | | | Tien Kung 1.0 | | |
| T1 | TK1-FindTape | 446 | 793 | Find the packaging tape, pick it up, and place it into another basket | |
| T2 | TK1-StackBowls | 260 | 497 | Put the blue bowl in the middle of the table, then stack the green bowl on top of it | |
| T3 | TK1-CloseDrawer | 231 | 223 | Slide the drawer closed | |
| T4 | TK1-FlipTennisTube | 300 | 535 | Put the tennis tube upright | |
| T5 | TK1-PressCookerButton | 98 | 187 | Press the rice cooker's off button | |
| T6 | TK1-MoveChopstickCup | 274 | 574 | Move the blue cup to the middle, then place one chopstick from the bamboo holder into it | |
| T7 | TK1-StackCubes | 200 | 520 | Stack the two blue cubes | |
| T8 | TK1-StackCups | 200 | 487 | Move the blue cup and stack it with the other blue cup | |
| T9 | TK1-StackPlates | 200 | 449 | Place the pink plate into the beige plate in the middle, then stack the blue plate on top of the pink plate | |
| T10 | TK1-PickWipeTowel | 222 | 643 | Pick up a towel and wipe the water with it | |

| # | Task | Trajectory | | Task Instruction | Task Setting |
|---|------|------------|---|------------------|--------------|
| | | Num. | Avg. Len. | | |
| T11 | TK1-HangTowel | 242 | 620 | Pick up the towel with the right arm, hand it over to the left arm, and hang it on the rack with the left arm |  |
| T12 | TK1-OpenPotLid | 104 | 486 | Open the pot lid |  |
| T13 | TK1-OpenOven | 21 | 228 | Open the oven |  |
| T14 | TK1-PackEggBox | 192 | 315 | Put the egg into the box and close the lid |  |
| T15 | TK1-CloseLaptop | 174 | 188 | Close the laptop screen |  |
| T16 | TK1-InserToaster | 174 | 254 | Insert the bread into the toaster |  |
| T17 | TK1-FlipCup | 153 | 578 | Flip the cup upright |  |
| T18 | TK1-PlaceFlipButton | 125 | 616 | Pick up the button with the right arm and place it in the middle. Then use the left arm to flip the button upright |  |
| T19 | TK1-OpenTrashBin | 177 | 206 | Open the trash bin |  |
| T20 | TK1-PressMachine | 181 | 213 | Press down the bread machine with the right arm |  |
| **Dual-Arm Franka** | | | | | |
| T1 | DFR-MoveCupMilk | 293 | 254 | Place the cup in the middle of the table, then pick up the milk and put it next to the cup |  |
| T2 | DFR-StackBowls | 298 | 245 | Put the blue bowl in the middle of the table and stack the green bowl on top of it |  |

| # | Task | Trajectory | | Task Instruction | Task Setting |
|---|------|------|------|---|---|
| | | Num. | Avg. Len. | | |
| T3 | DFR-SweepTrash | 248 | 350 | Sweep up the rubbish and take out the trash |  |
| T4 | DFR-TransferCup | 244 | 196 | Pick up the cup with the right arm, hand it over to the left arm, and hang it on the holder with the left arm |  |
| T5 | DFR-MoveChopstick | 277 | 299 | The left arm moves the blue cup from the left side of the robot to the middle, while the right arm takes a chopstick from the bamboo holder on the right side and puts it into the blue cup |  |
| T6 | DFR-StackCubes | 245 | 166 | Put the blue cube in the middle of the desk and stack it on top of the other blue one |  |
| T7 | DFR-StackPlates | 360 | 230 | Use the left arm to place the pink plate into the beige plate in the middle, then use the right arm to stack the blue plate on top of the pink plate |  |
| T8 | DFR-CleanTable | 284 | 201 | Put the trash into the trash can, and put the items back in the box |  |
| T9 | DFR-HangCupHolder | 206 | 199 | Hang the cup on the cup holder |  |
| T10 | DFR-HangTowelRack | 232 | 195 | Pick up the towel with the right arm, hand it over to the left arm, and hang it on the rack with the left arm |  |
| T11 | DFR-FindTapeBox | 194 | 245 | Find the packaging tape and put it into the other box |  |
| T12 | DFR-PickButtonPress | 200 | 206 | The left arm picks up the green button and places it in the middle, while the right arm presses it |  |
| T13 | DFR-SweepRubbish | 196 | 288 | Sweep up the rubbish |  |
| T14 | DFR-CloseToolbox | 201 | 220 | Use both arms to close the toolbox |  |

| # | Task | Trajectory | | Task Instruction | Task Setting |
|---|------|------|------|---|---|
| | | Num. | Avg. Len. | | |
| T15 | DFR-CollectBasketTea | 200 | 186 | The right arm places the shopping basket in the middle, while the left arm takes tea drinks from the shelf and puts them inside |  |
| T16 | DFR-PlaceTools | 199 | 159 | Use the right arm to place the wrench on the right side of the toolbox, and use the left arm to place the screwdriver on the left side |  |
| T17 | DFR-GetBlocks | 94 | 255 | The right arm grabs the storage box and opens the lid, while the left arm places the red building blocks inside, ensuring they do not fall off |  |
| T18 | DFR-PlaceRagWipe | 196 | 301 | Use the right arm to place the rag in the middle of the table, and use the left arm to wipe the remaining liquid on the middle of the table with the rag |  |
| T19 | DFR-OpenToolbox | 199 | 244 | Use both arms to open the toolbox |  |
| T20 | DFR-PlaceScrews | 189 | 301 | The right arm places the two long screws into the slot at the very right end of the storage box, while the left arm places the two short screws into the slot at the very left end of the storage box |  |
| **AgileX Cobot Magic V2.0** | | | | | |
| T1 | AGX-OpenDrawerButton | 272 | 1418 | Slide open the drawer and place the yellow button inside |  |
| T2 | AGX-MoveButtonDrawer | 387 | 1612 | Place the yellow button in the drawer and close it |  |
| T3 | AGX-StackBoxes | 169 | 1514 | Put the left box in the middle, then stack the right box on top of it |  |
| T4 | AGX-FindTapeBox | 182 | 1058 | Find the packaging tape and put it into the other box |  |
| T5 | AGX-SweepRubbish | 112 | 2370 | Sweep up the rubbish |  |

| # | Task | Trajectory | | Task Instruction | Task Setting |
|---|---|---|---|---|---|
| | | Num. | Avg. Len. | | |
| T6 | AGX-ArrangeValves | 194 | 1411 | Arrange the valves in a row | |
| T7 | AGX-HangScissors | 48 | 3124 | Hang the scissors on the holder | |
| T8 | AGX-PlaceButton | 184 | 1794 | Take the blue tray and place a button on it | |
| T9 | AGX-CloseToolbox | 184 | 2003 | Use both arms to close the toolbox | |
| T10 | AGX-GatherScrews | 152 | 2918 | The right arm places the two long screws into the slot at the very right end of the storage box, while the left arm places the two short screws into the slot at the very left end of the storage box | |
| T11 | AGX-FindCircuit | 476 | 2880 | The left arm picks up the circuit breaker from the red tray and places it in the middle of the table. Then the right arm picks up the circuit breaker and puts it into the blue tray on the right | |
| T12 | AGX-PlaceBiscuitBox | 188 | 1983 | Pick up the biscuit box from the blue basket with the right arm and place it in the middle of the table. Then, use the left arm to place it on the middle shelf of the black rack | |
| T13 | AGX-CollectBasketTea | 190 | 3078 | The right arm places the shopping basket in the middle, while the left arm takes tea drinks from the shelf and puts them inside | |
| T14 | AGX-PlaceScrewdriver | 177 | 3079 | The right arm picks up the Phillips screwdriver and places it in the middle of the table. Then, the left arm picks it up again and puts it into the groove in the toolbox | |
| T15 | AGX-PourGearOil | 192 | 3144 | The left arm takes the gear and places it on the middle metal tray, and the right arm pours lubricating oil on the gear | |
| T16 | AGX-StackBrakePads | 188 | 1650 | Use the left arm to place Brake Pad Type A in the middle, and use the right arm to pick up Brake Pad Type B and stack it on top of Brake Pad Type A | |

| # | Task | Trajectory | | Task Instruction | Task Setting |
|---|------|-----|------|------------------|--------------|
| | | Num. | Avg. Len. | | |
| T17 | AGX-MeshStackCup | 117 | 1765 | Place the mesh and stack the cup on it |  |
| T18 | AGX-PourWine | 162 | 1644 | Pour the wine with the right arm and place the cup on the tray with the left arm |  |
| T19 | AGX-HangWipeRag | 199 | 2257 | Use the right arm to place the rag in the middle of the table, and use the left arm to wipe the remaining liquid with it |  |
| T20 | AGX-StackBowls | 185 | 1314 | Stack the blue bowl on top of the green bowl |  |
| | | **Single-Arm UR5e** | | | |
| T1 | SUR-FindTape | 134 | 104 | Find the packaging tape and put it into the other basket |  |
| T2 | SUR-MoveMilkCup | 292 | 116 | Pick up the milk and place it next to the cup |  |
| T3 | SUR-StackBowls | 300 | 118 | Stack the blue bowl on top of the green bowl |  |
| T4 | SUR-OpenDrawer | 212 | 168 | Slide open the drawer |  |
| T5 | SUR-CloseDrawer | 190 | 191 | Slide the drawer closed |  |
| T6 | SUR-InsertToyBlock | 150 | 198 | Insert the blue toy into the square-bottom slot of the grey block |  |

| # | Task | Trajectory | | Task Instruction | Task Setting |
|---|------|------|------|---|---|
| | | Num. | Avg. Len. | | |
| T7 | SUR-PlaceChopstick | 297 | 109 | Place one chopstick from the bamboo chopstick holder into the blue cup |  |
| T8 | SUR-StackCubes | 308 | 92 | Stack the two blue cubes on top of each other |  |
| T9 | SUR-StackCup | 284 | 192 | Stack the cups |  |
| T10 | SUR-StackPlates | 291 | 194 | Stack the plates in the middle |  |
| T11 | SUR-SlideDrawer | 56 | 181 | Slide open the drawer |  |
| T12 | SUR-OpenUpperDrawer | 182 | 132 | Open the upper drawer |  |
| T13 | SUR-OpenOven | 141 | 70 | Open the oven |  |
| T14 | SUR-PackEggBox | 183 | 151 | Put the egg into the box and close the lid |  |
| T15 | SUR-CloseLaptop | 196 | 107 | Close the laptop screen |  |
| T16 | SUR-InsertBread | 193 | 156 | Insert the bread into the toaster |  |

| # | Task | Trajectory | | Task Instruction | Task Setting |
|---|---|---|---|---|---|
| | | Num. | Avg. Len. | | |
| T17 | SUR-AssembleValve | 182 | 166 | Assemble the valve |  |
| T18 | SUR-PourTubeBeaker | 152 | 270 | Pick up the test tube and pour water into a 50 ml glass beaker |  |
| T19 | SUR-PourGearOil | 172 | 214 | Take the gear and place it on the middle metal tray, then pour lubricating oil on it |  |
| T20 | SUR-WipeHangRag | 209 | 134 | After using the rag to wipe the water in the middle of the table, hang the rag on the rag rack |  |

