# OpenReview forum: "XR-1: Towards Versatile Vision-Language-Action Models via Learning Unified Vision-Motion Representations"
_ICLR.cc/2026/Conference — Submitted to ICLR 2026_

### Official Review · Reviewer_gGq7 · 2025-10-29

**Soundness:** 4
**Presentation:** 4
**Contribution:** 4
**Rating:** 10
**Confidence:** 5

**Summary:**

To bridge gaps across heterogeneous data sources, this paper proposes a novel framework, XR-1, for versatile and scalable VLA learning. The core of XR-1 is the Unified Vision-Motion Codes (UVMC) which jointly encodes visual dynamics and robotic motion via a dual-branch VQ-VAE. Through UVMC, the vision dynamic information and motion action are aligned and can be serving as an intermediate representation. Based on UVMC, a three-stage training paradigm is proposed: (1) self-supervised UVMC learning, (2) UVMC-based pretaining on large-scale cross-embodiment robotic datasets, and (3) task-specific post-training. Extensive experiments verify the superior performance over SoTAs such pi-0. Ablation study is also well designed to show the effectiveness of the proposed training paradigm.

**Strengths:**

1. This paper is well-presented and easy to follow. The motivations are clear and insightful, and the methodology is straightforward to understand.

2. The idea is both novel and interesting. Achieving generalization across different robots and learning from heterogeneous data is an important challenge. The unified representation method that aligns visual dynamics with action execution based on UVMC is an inspiring approach to address this issue. While similar training paradigms can be found in other computer vision tasks, such as research on text-to-human motion, this represents the first application of such a paradigm in robot action learning, with adaptations made to fit the robotics domain.

3. Comprehensive experiments effectively demonstrate the superior performance of the proposed method, and the ablation studies validate the effectiveness of the designed components and the training paradigm.

**Weaknesses:**

1. Generally, self-supervised training on VQ-VAE is a crucial stage, and the learning of the codebook significantly impacts performance. However, this paper appears to lack discussion on this aspect, and certain details, such as the size of the codebook, remain unspecified, which may affect reproducibility.

2. In the proposed method, human demonstration videos (Ego4D) are utilized during the pretraining stage, showcasing its capability to leverage heterogeneous data sources and cross-embodiment. This is a significant strength. However, the impact of using human source data has not been analyzed.

3. Typos, for instance,  $z_{mo}^e$ after "visual dynamics codes" should be "$z_{vis}^e$" in line 229.

**Questions:**

1. In stage 2 and 3, how are actions decoded? Using the pretrained VQ-VAE decoder or a new action head?

---

> ### Author Response · Authors · 2025-11-25
> **Response to Reviewer gGq7 (Part 1)**
>
> We thank the reviewer for the insightful and constructive comments. We denote Weakness $n$ of Reviewer gGq7 as R4Wn and Question $n$ as R4Qn.
>
> **[R4W1]**
> > More ablation studies of UVMC.
>
> We appreciate the constructive suggestion to investigate the impact of different codebook variants. To comprehensively address this concern, along with related points raised by Reviewer vfX7 (Weakness 1) and Reviewer H8iB (Weakness 4), we conducted extensive ablation studies on the Dual-Arm UR-5e embodiment across six tasks. As presented in **Table R4_1**, our experiments evaluated the performance impact of using a separate codebook structure as well as restricting the latent representation to single-modality (vision-only or motion-only) codes.
>
> The results indicate that employing a separate codebook yields performance comparable to our shared codebook baseline. We attribute this to the alignment loss, which effectively constrains the relationship between codes of different modalities.
> In contrast, relying solely on either vision or motion codes leads to a significant drop in success rates, validating that the synergy between both modalities is crucial for final performance. We are currently finalizing additional experiments regarding codebook size and will update these results shortly. All findings from these ablation studies will be integrated into the appendix of the revision.
>
> Table R4_1: Success rate results across 6 tasks on Dual-Arm UR-5e. "DT" indicates training directly on the downstream task data.
> | sepe codebook | latent token | stage-1 | stage2 | stage3 | DUR-CleanTable | DUR-FindTapeBasket | DUR-MoveCupMilk | DUR-StackBowls | DUR-SweepTrash | DUR-TransCupHolder | Avg.  |   |
> |---------------|--------------|---------|--------|--------|----------------|--------------------|-----------------|----------------|----------------|--------------------|-------|---|
> | T             | both         | DT      | DT     | DT     | 55             | 75                 | 55              | 85             | 40             | 80                 | 65.0    |   |
> | F             | motion       | DT      | DT     | DT     | 20             | 35                 | 35              | 55             | 15             | 25                 | 30.8  |   |
> | F             | vision       | DT      | DT     | DT     | 30             | 40                 | 45              | 70             | 55             | 60                 | 50.0    |   |
> | F             | both         | DT      | DT     | DT     | 50             | 75                 | 65              | 80             | 60             | 70                 | 66.7  |   |
>
>
> **[R4W2]**
>
> > Impact of the human video data (Ego4D).
>
> Thanks for your constructive suggestion. The full Ego4D dataset contains 14.3M frames, accounting for 8.7% of the Stage 1 data.
> Due to the large scale and our computational constraints, we have set up a comparison where we reduce the Stage 1 pre-training dataset by a factor of 10 and evaluate the effect of including versus excluding Ego4D.
> The results are shown in the **Table R4_2** below: when Ego4D is removed (w/o Ego4D), the average success rate decreases by 5.8% compared with the setting that includes Ego4D, quantitatively demonstrating the positive impact of Ego4D on pre-training.
> We will integrate these comprehensive ablation studies into the appendix of the revision.
>
> Please also note that quantifying the exact performance gain from Ego4D is not the core contribution of this paper. Our main focus is that XR-1 provides a way to leverage human video data for learning; as more human video becomes available in the future, XR-1 has a higher performance ceiling and greater potential than VLA training paradigms that rely solely on action-labeled data.
>
> Table R4_2: Comparison of pre-training results using Ego4D human video data on 6 dual-arm UR tasks..
> | Instantiation  | Stage-1 PT. | Stage-2 | Stage-3 | DUR-CleanTable | DUR-FindTapeBasket | DUR-MoveCupMilk | DUR-StackBowls | DUR-SweepTrash | DUR-TransCupHolder | Avg  |
> |----------------|------------:|--------:|--------:|---------------:|-------------------:|----------------:|--------------:|--------------:|-------------------:|-----:|
> | XR-1 w/o Ego4D | 10%         | Downstream Data       | ✓       | 20             | 60                 | 10              | 55            | 15            | 35                 | 32.5 |
> | XR-1 w/ Ego4D  | 10%         | Downstream Data       | ✓       | 25             | 60                 | 25              | 60            | 20            | 40                 | **38.3** |

---

> ### Author Response · Authors · 2025-11-25
> **Response to Reviewer gGq7 (Part 2)**
>
> **[R4W3]**
> > Typos and notation errors.
>
> We appreciate the reviewer's attention to detail. We have corrected the notation error in Line 229. Furthermore, we have thoroughly proofread the entire manuscript to ensure all typos and notation inconsistencies are resolved in the revision.
>
>
> **[R4Q1]**
>
> > In stages 2 and 3, how are actions decoded? Using the pretrained VQ-VAE decoder or a new action head?
>
> We appreciate the reviewer's query regarding the decoding mechanism. The XR-1 framework is designed to be model-agnostic, supporting both diffusion-based and autoregressive VLA instances. In our implementation, we built two diffusion-based variants: XR-1 (built on the $\pi_0$ architecture) and XR-1-Light (built on Switch-VLA). Both models utilize a diffusion-based action head that generates precise control signals by iteratively denoising noisy action inputs.
>
> Regarding the specific decoder architecture used in Stages 2 and 3:
> For the pi0-based XR-1, we reuse the pretrained VQ-VAE decoder (the action expert network established in Stage 1) for Stages 2 and 3.
> In contrast, for the XR-1-Light, due to the change in backbone architecture, we employ a new action head following the Switch-VLA.

---

### Official Review · Reviewer_vfX7 · 2025-10-31

**Soundness:** 3
**Presentation:** 3
**Contribution:** 2
**Rating:** 6
**Confidence:** 4

**Summary:**

This paper introduces XR-1, a framework for building generalist Vision-Language-Action (VLA) robots. Its core contribution is the Unified Vision-Motion Codes (UVMC), a discrete latent representation that jointly encodes visual dynamics and robotic actions into a shared space via a dual-branch VQ-VAE. Through a three-stage training paradigm, XR-1 leverages this unified representation to enable precise low-level control and strong cross-embodiment generalization, demonstrably outperforming state-of-the-art models across 120+ tasks on six different robot embodiments.

**Strengths:**

1. The technical quality and empirical rigor are exceptional. The paper is grounded by an immense evaluation campaign.
2. The consistent and significant outperformance of strong, recent baselines (e.g., $\pi_{0}$, GR00T-N1.5) across diverse scenarios (bimanual, dexterous, long-horizon) provides compelling evidence for the framework's effectiveness.

**Weaknesses:**

1. For a fairer assessment of UVMC's value, the authors should include baselines that use alternative representation learning paradigms. For instance, a comparison with methods that use only visual dynamics latents or only action latents would directly isolate the benefit of the unified vision-motion approach.
2. The paper treats the UVMC as a "black box." While the results demonstrate its effectiveness, it lacks a qualitative or quantitative analysis of what the model learns in this unified space. The authors may consider the following: (1) Perform a nearest-neighbor analysis in the codebook space. For example, show that the motion code for "grasp" is close to the visual code for a hand closing around an object, even across different embodiments; (2) Visualize the latent space with t-SNE/UMAP plots, coloring points by task semantics (e.g., "picking," "placing") versus embodiment-specific details, to empirically demonstrate the embodiment-agnostic and semantically structured nature of UVMC.
3. The contribution of the large-scale human video data (Ego4D) is unclear and not quantitatively ablated. The paper states human data uses only $L_{vis}$ , but it's ambiguous how these *actionless* videos ultimately improve low-level robot control.
4. The paper extensively reports success rates but provides little discussion of *when and why XR-1 fails*. Understanding the limitations is crucial for future research.

**Questions:**

All my questions are highly related to the weaknesses.

1.  To isolate UVMC's value, could you include baselines that use only visual or only action latents, demonstrating the specific benefit of their unification?

2.  Could you provide an analysis (e.g., nearest-neighbor retrieval or latent space visualization) to show that UVMC learns semantically meaningful, embodiment-agnostic clusters?

3.  What is the quantitative impact of the human video data (Ego4D) on final robot performance? An ablation study removing it from Stage-1 would clarify its contribution.

4.  Could you discuss the primary failure modes of XR-1? In which task types does it consistently struggle, and what is the typical root cause (perception, UVMC prediction, or control)?

---

> ### Author Response · Authors · 2025-11-25
> **Response to Reviewer vfX7 (Part 1)**
>
> We thank the reviewer for the insightful and constructive comments. We denote Weakness $n$ of Reviewer vfX7 as R3W$n$ and Question $n$ as R3Q$n$.
>
> **[R3W1]**
> > Baselines that use alternative representation learning paradigms (only visual dynamics latents or only action latents).
>
> We appreciate the constructive suggestion to investigate the impact of using only vision codes or only motion codes.
> To address this, we conducted an ablation study comparing these variants.
> As presented in **Table R3_1**, we compared the success rates of both variants across 6 tasks on the Dual-Arm UR-5e embodiment.
> The results demonstrate that relying solely on either vision or motion codes leads to a significant drop in success rates, thereby validating the crucial contribution of both modalities to the final performance.
> Furthermore, in response to Reviewer H8iB's fourth weakness comment, we have conducted additional ablation studies to evaluate the impact of using a separate codebook.
> We will integrate these comprehensive ablation studies into the appendix of the revision.
>
> Table R3_1: Success rate results across 6 tasks on Dual-Arm UR-5e. "DT" indicates training directly on the downstream task data.
> | sepe codebook | latent token | stage-1 | stage2 | stage3 | DUR-CleanTable | DUR-FindTapeBasket | DUR-MoveCupMilk | DUR-StackBowls | DUR-SweepTrash | DUR-TransCupHolder | Avg.  |   |
> |---------------|--------------|---------|--------|--------|----------------|--------------------|-----------------|----------------|----------------|--------------------|-------|---|
> | T             | both         | DT      | DT     | DT     | 55             | 75                 | 55              | 85             | 40             | 80                 | 65.0    |   |
> | F             | **motion**       | DT      | DT     | DT     | 20             | 35                 | 35              | 55             | 15             | 25                 | 30.8  |   |
> | F             | **vision**       | DT      | DT     | DT     | 30             | 40                 | 45              | 70             | 55             | 60                 | 50.0    |   |
> | F             | both         | DT      | DT     | DT     | 50             | 75                 | 65              | 80             | 60             | 70                 | 66.7  |   |

---

> ### Author Response · Authors · 2025-11-25
> **Response to Reviewer vfX7 (Part 2)**
>
> **[R3W2]**
> > More analysis of the unified space.
>
> We appreciate the reviewer’s constructive suggestion regarding the interpretability of UVMC alignment. To address this, we have conducted additional visualization experiments and included the detailed results in **Appendix 6.7** of the revised paper.
>
> To assess whether UVMC learns a semantically aligned latent space between actions and visual observations, we conduct a nearest-neighbor retrieval experiment between motion and vision codes (Figure 11). We select 7 tasks, each with 10 trajectory episodes, compute VC and MC for all frames, and at time step T extract the MCs for GRASP and MOVE and visualize the corresponding frames. For each such MC, we compute VC features from different tasks and embodiments and, using cosine similarity, retrieve its nearest and farthest VC neighbors under four settings: Same Embodiment & Similar Task (SE.ST.), Same Embodiment & Different Task (SE.DT.), Different Embodiment & Similar Task (DE.ST.), and Different Embodiment & Different Task (DE.DT.). For the different-embodiment setting, we use a dual-arm Franka and standardize left/right with respect to the embodiment’s outward-facing direction. The key observations from Figure 11 are:
> 1. SE.ST. (Same embodiment, similar task):
> Nearest-neighbor VCs closely match the GRASP and MOVE semantics of the query MCs, while farthest-neighbor images show clearly different motions.
> 2. SE.DT. (Same embodiment, different task):
> Even when the task changes, nearest-neighbor VCs still correspond to grasp or move, whereas farthest-neighbor images correspond to distinct, non-matching motions.
> 3. DE.ST. (Different embodiment, similar task):
> Across different embodiments, nearest-neighbor frames under similar tasks still depict similar actions, in contrast to the clearly different motions in the farthest-neighbor frames.
> 4. DE.DT. (Different embodiment, different task):
> Despite changes in both embodiment and task, nearest-neighbor retrieval continues to select frames whose semantics are closest to grasp or move, while farthest-neighbor images represent dissimilar actions.
> Together, these results in Figure 11 show that UVMC learns meaningful action semantics in its latent space and that these semantic representations are embodiment-agnostic.
>
> To qualitatively validate whether UVMC effectively captures intrinsic task dynamics within trajectory distributions, we employed t-SNE to project the high-dimensional latent embeddings into a 2D manifold. We analyzed two distinct scenarios: (1) a single-robot setting involving a dual-arm UR robot performing 6 tasks, and (2) a mixed-embodiment setting comprising both dual-arm Franka and UR robots. As illustrated in the figures, the results demonstrate that UVMC maintains strong **Semantic Consistency of Dynamics**, effectively disentangling common patterns (e.g., picking) from task-specific nuances (e.g., sweeping). Furthermore, the visualizations confirm robust **Cross-Embodiment Alignment**: the model successfully aggregates feature embeddings for similar tasks across different robot configurations. This indicates that UVMC learns an embodiment-invariant representation that prioritizes high-level task semantics over low-level proprioceptive differences.

---

> ### Author Response · Authors · 2025-11-25
> **Response to Reviewer vfX7 (Part 3)**
>
> **[R3W3]**
> > Impact of the human video data (Ego4D).
>
> Thanks for your constructive suggestion. The full Ego4D dataset contains 14.3M frames, accounting for 8.7% of the Stage 1 data.
> Due to the large scale and our computational constraints, we have set up a comparison where we reduce the Stage 1 pre-training dataset by a factor of 10 and evaluate the effect of including versus excluding Ego4D.
> The results are shown in the **Table R3_2** below: when Ego4D is removed (w/o Ego4D), the average success rate decreases by 5.8% compared with the setting that includes Ego4D, quantitatively demonstrating the positive impact of Ego4D on pre-training.
> We will integrate these comprehensive ablation studies into the appendix of the revision.
>
> Please also note that quantifying the exact performance gain from Ego4D is not the core contribution of this paper. Our main focus is that XR-1 provides a way to leverage human video data for learning; as more human video becomes available in the future, XR-1 has a higher performance ceiling and greater potential than VLA training paradigms that rely solely on action-labeled data.
>
> Table R3_2: Comparison of pre-training results using Ego4D human video data on 6 dual-arm UR tasks.
> | Instantiation  | Stage-1 PT. | Stage-2 | Stage-3 | DUR-CleanTable | DUR-FindTapeBasket | DUR-MoveCupMilk | DUR-StackBowls | DUR-SweepTrash | DUR-TransCupHolder | Avg  |
> |----------------|------------:|--------:|--------:|---------------:|-------------------:|----------------:|--------------:|--------------:|-------------------:|-----:|
> | XR-1 w/o Ego4D | 10%         | Downstream Data       | ✓       | 20             | 60                 | 10              | 55            | 15            | 35                 | 32.5 |
> | XR-1 w/ Ego4D  | 10%         | Downstream Data       | ✓       | 25             | 60                 | 25              | 60            | 20            | 40                 | **38.3** |
>
> **[R3W4]**
> > Failure case analysis on XR-1.
>
> We conducted a qualitative review of the rollout videos to investigate why XR-1 fails and added the analysis to Appendix 6.9 (highlighted in red in the updated pdf). We found the failures belong to the primary mode below:
>
> 1. Precision Deficiency.
> The most common failure mode involves the robot attempting the correct action but failing in execution precision or bimanual coordination.
> Example: In the `TK2-CollectScrews` task, the robot may fail to grasp a tiny screw or drop it mid-motion due to slight localization errors. This reflects the inherent difficulty of learning precise, bimanual coordination for dexterous manipulation tasks.

---

### Official Review · Reviewer_PrUm · 2025-10-31

**Soundness:** 1
**Presentation:** 3
**Contribution:** 2
**Rating:** 2
**Confidence:** 4

**Summary:**

This paper presents XR-1, a unified vision-language-action framework that introduces the Unified Vision-Motion Codes (UVMC) to jointly quantize and align visual dynamics with motor actions. The authors conduct extensive real-world experiments with over 12,000 rollouts across six different robot embodiments, demonstrating strong cross-task and cross-embodiment generalization.

**Strengths:**

- The paper presents extensive real-world experiments, validating XR-1 with over 12,000 rollouts across six different robot embodiments.
- This paper presents XR-1, a unified vision-language-action framework that introduces the Unified Vision-Motion Codes (UVMC) to jointly quantize and align visual dynamics with motor actions.
The authors conduct extensive real-world experiments with over 12,000 rollouts across six different robot embodiments, demonstrating strong cross-task and cross-embodiment generalization.The proposed method is novel, introducing the Unified Vision-Motion Codes (UVMC) that jointly quantize and encode both actions and observations.

**Weaknesses:**

- The paper does not appear to include simulation experiments. Although the real-world evaluation is rich and convincing, an evaluation on a standard public benchmark would provide a more comprehensive assessment of performance and facilitate reproducibility for the research community. *I strongly suggest author take this into consideration*.
- Analytical justification of UVMC: The core innovation UVMC lacks deeper justification beyond ablation studies. The observed improvements could potentially be influenced by hyperparameter choices or implementation factors. I recommend adding more analytical or theoretical explanations to strengthen the evidence for UVMC’s effectiveness.
- Interpretability of UVMC: The interpretability of the UVMC alignment mechanism could also be improved. Although the KL alignment encourages visual and motion features to share the same codebook, the paper does not analyze how these codes are organized or clustered in semantic space. For example, which codes may correspond to grasping, rotation, or translation. As a result, the semantic consistency of UVMC remains an assumption rather than an explicitly demonstrated property.

**Questions:**

- I am somewhat unclear about the UVMC design. During reconstruction, both observation and action encoders take additional inputs such as c_t and l, o. Could the authors provide more explanation on the rationale behind this design?
- The latent variables are defined as $z_{\text{vis}} = E_{\text{vis}}(c_t, c_{t+h})$ and $z_{\text{mo}} = E_{\text{mo}}(a_{t:t+h}, m_{t:t+h}).$
This paper propose to let the latent representation be as close as possible. However, therotically, the same $(a_{t:t+h}, m_{t:t+h})$ can correspond to very different $(c_t, c_{t+h})$, since the action sequence and motor states only describe the robot itself, while the visual inputs capture the entire scene, including background and objects, which can be changed arbitrarily across episodes. Directly aligning these two spaces might only be reasonable under specific dataset distributions. Could the authors clarify this assumption or provide further justification?

---

> ### Author Response · Authors · 2025-11-25
> **Response to Reviewer PrUm (Part 1)**
>
> We thank the reviewer for the insightful and constructive comments. We denote Weakness $n$ of Reviewer PrUm as R2Wn and Question $n$ as R2Qn.
>
> **[R2W1]**
> > Comprehensive Assessment and Reproducibility.
>
> We thank the reviewer for the constructive suggestions. We respectfully submit that our paper already presents a comprehensive assessment and ensures reproducibility, primarily driven by our extensive focus on physical world performance. We outline our justification and additional experimental results below:
>
> 1. Large Scale of Real-Robot Evaluation. To the best of our knowledge, we have established the largest-scale real-robot evaluation experiments to date. Our assessment encompasses **over 14,000 rollouts across 6 different robot embodiments and 120+ tasks** (we have also included additional evaluation results for $\pi_{0.5}$[1] in the updated PDF). This scale significantly exceeds that of recent leading works, such as $\pi_0$ (20 tasks with 660 rollouts), RDT (7 tasks with 875 rollouts), and GR00T (13 tasks with 520 rollouts). We believe this extensive coverage—spanning diverse hardware, task varieties, and evaluation volume—provides a robust and sufficient demonstration of our model's effectiveness in the physical world.
>
> 2. Focus on Real-World Applicability. We maintain that the ultimate goal of embodied AI is deployment in physical environments; consequently, large-scale real-world evaluation is often more convincing and challenging than simulation. Recent influential works, such as RDT[2], $\pi_0$[3], and $\pi_{0.5}$[1], have similarly focused exclusively on real-world validation without relying on simulation benchmarks, yet have made significant contributions to the community.
>
> 3. Commitment to Reproducibility. To fully address concerns regarding reproducibility, we are committed to open-sourcing both our code and model checkpoints. This will ensure the research community can validate our findings and build upon our work.
>
>
> 4. Supplementary Simulation Results. As recommended, we conducted additional preliminary experiments on the SimplerEnv Benchmark (WidowX, 4 tasks) to compare our model against $\pi_0$. As shown in **Table R2_1**, XR-1 achieves an average success rate of 0.503, outperforming $\pi_0$'s 0.396.
> While the absolute success rates for both models suggest room for improvement, we attribute this primarily to the significant domain gap between simulation and the real world. Furthermore, these results typically require specific hyperparameter fine-tuning for simulation environments to achieve optimal scores.
> It is important to note that we retained the hyperparameters optimized for real-world robot deployment during these tests.
> Even under this challenging setting without simulation-specific tuning, XR-1 demonstrates superior robustness and performance compared to the baseline.
>
> Table R2_1: Success rate results on SimplerEnv-WidowX across 4 tasks.
> | Method  | Put Spoon | Put Carrot | Stack Block | Put Eggplant | Average |
> |---------|-----------|------------|-------------|--------------|---------|
> | $\pi_0$ | 0.33      | 0.42       | 0.29        | 0.54         | 0.396   |
> | XR-1    | 0.54      | 0.63       | 0.38        | 0.46         | 0.503   |
>
> [1] Intelligence, Physical, et al. "$\pi_ {0.5} $: a Vision-Language-Action Model with Open-World Generalization." arXiv preprint arXiv:2504.16054 (2025).
>
> [2] Liu, Songming, et al. "Rdt-1b: a diffusion foundation model for bimanual manipulation." Proceedings of the International Conference on Learning Representations (ICLR) (2025).
>
> [3] Black, Kevin, et al. "$\pi_0 $: A Vision-Language-Action Flow Model for General Robot Control." Proceedings of Robotics: Science and Systems (RSS) (2025).

---

> ### Author Response · Authors · 2025-11-25
> **Response to Reviewer PrUm (Part 2)**
>
> **[R2W2]**
> > Analytical justification of UVMC.
>
> We appreciate the reviewer’s insightful suggestion for a deeper analytical justification of the Unified Vision-Motion Codes (UVMC). Beyond the empirical improvements observed in our ablation studies, the effectiveness of UVMC is grounded in two fundamental theoretical principles: **Cross-Modal Manifold Alignment** and **Information Bottleneck Regularization**.
>
> 1. From a manifold alignment perspective, high-dimensional visual observations and low-dimensional motor commands can be viewed as two distinct projections of the same underlying physical state transition. Existing VLA approaches force the model to learn a complex, highly non-linear mapping directly between these disparate modalities. UVMC simplifies this optimization problem by establishing a shared, discrete latent space. The dual-branch VQ-VAE, constrained by our cross-modality alignment loss, forces the visual embedding $z_{vis}$ and the motion embedding $z_{mo}$ to map onto a common low-dimensional manifold. This explicitly encodes the internal link between visual dynamics and robotic action, ensuring that the visual representation is structurally isomorphic to the control space. Consequently, the VLM backbone is relieved of the burden of implicit alignment, allowing it to focus on precies action prediction.
>
> 2. Furthermore, UVMC functions as a critical information bottleneck. Raw visual inputs contain a vast amount of task-irrelevant high-frequency information (e.g., lighting variations, dynamic backgrounds, texture shifts). By discretizing these inputs into a limited codebook shared with motor actions, UVMC acts as a filter that discards visual "nuisance variables" that are not predictive of physical motion. This results in a representation that is robust to environmental noise and focuses exclusively on action-relevant dynamics. This design is not merely a hyperparameter optimization but a structural inductive bias that aligns with the "supramodal" cognitive processing[1] observed in humans, where sensory details are abstracted into latent representation.
>
> 3. Finally, regarding the concern about implementation factors, we emphasize that the performance gains provided by UVMC are consistent across significantly different model architectures (the large-scale XR-1 based on PaliGemma and the compact XR-1-Light based on SwitchVLA) and across diverse robotic embodiments (Dual-Arm UR-5e, Franka, Tien Kung). As shown in Table 4, the removal of UVMC results in a catastrophic performance drop (from 66.7% in Exp.5 to 28.3% in Exp.3), a magnitude of difference that far exceeds variance typically attributable to hyperparameter tuning. This consistency confirms that UVMC provides a fundamental representational advantage necessary for precise low-level control in generalist policies.
>
> [1] Park, Doyoung, et al. "Supramodal and cross-modal representations of working memory in higher-order cortex." Nature Communications 16.1 (2025): 4497.

---

> ### Author Response · Authors · 2025-11-25
> **Response to Reviewer PrUm (Part 3)**
>
> **[R2W3]**
> > Interpretability of UVMC.
>
> We appreciate the reviewer’s constructive suggestion regarding the interpretability of UVMC alignment. To address this, we have conducted additional visualization experiments and included the detailed results in **Appendix 6.7** of the revised paper.
>
> To assess whether UVMC learns a semantically aligned latent space between actions and visual observations, we conduct a nearest-neighbor retrieval experiment between motion and vision codes (Figure 11). We select 7 tasks, each with 10 trajectory episodes, compute VC and MC for all frames, and at time step T extract the MCs for GRASP and MOVE and visualize the corresponding frames. For each such MC, we compute VC features from different tasks and embodiments and, using cosine similarity, retrieve its nearest and farthest VC neighbors under four settings: Same Embodiment & Similar Task (SE.ST.), Same Embodiment & Different Task (SE.DT.), Different Embodiment & Similar Task (DE.ST.), and Different Embodiment & Different Task (DE.DT.). For the different-embodiment setting, we use a dual-arm Franka and standardize left/right with respect to the embodiment’s outward-facing direction. The key observations from Figure 11 are:
> 1. SE.ST. (Same embodiment, similar task):
> Nearest-neighbor VCs closely match the GRASP and MOVE semantics of the query MCs, while farthest-neighbor images show clearly different motions.
> 2. SE.DT. (Same embodiment, different task):
> Even when the task changes, nearest-neighbor VCs still correspond to grasp or move, whereas farthest-neighbor images correspond to distinct, non-matching motions.
> 3. DE.ST. (Different embodiment, similar task):
> Across different embodiments, nearest-neighbor frames under similar tasks still depict similar actions, in contrast to the clearly different motions in the farthest-neighbor frames.
> 4. DE.DT. (Different embodiment, different task):
> Despite changes in both embodiment and task, nearest-neighbor retrieval continues to select frames whose semantics are closest to grasp or move, while farthest-neighbor images represent dissimilar actions.
> Together, these results in Figure 11 show that UVMC learns meaningful action semantics in its latent space and that these semantic representations are embodiment-agnostic.
>
> To qualitatively validate whether UVMC effectively captures intrinsic task dynamics within trajectory distributions, we employed t-SNE to project the high-dimensional latent embeddings into a 2D manifold. We analyzed two distinct scenarios: (1) a single-robot setting involving a dual-arm UR robot performing 6 tasks, and (2) a mixed-embodiment setting comprising both dual-arm Franka and UR robots. As illustrated in the figures, the results demonstrate that UVMC maintains strong **Semantic Consistency of Dynamics**, effectively disentangling common patterns (e.g., picking) from task-specific nuances (e.g., sweeping). Furthermore, the visualizations confirm robust **Cross-Embodiment Alignment**: the model successfully aggregates feature embeddings for similar tasks across different robot configurations. This indicates that UVMC learns an embodiment-invariant representation that prioritizes high-level task semantics over low-level proprioceptive differences.

---

> ### Author Response · Authors · 2025-11-25
> **Response to Reviewer PrUm (Part 4)**
>
> **[R2Q1]**
> > Explanation about the UVMC design.
>
> We appreciate the reviewer’s insightful inquiry regarding the UVMC architecture. We interpret the "additional inputs" mentioned in the review as referring to the decoder during the reconstruction phase, rather than inputs to the encoder. Both branches of our proposed framework adopt an asymmetric VQ-VAE paradigm [1], wherein the decoder is explicitly conditioned on specific contextual information to facilitate effective target reconstruction. We have updated Section 3.2 (highlighted in red) to provide a more detailed explanation of these architectural choices.
>
> 1. Vision Branch: The vision encoder takes both the current frame $c_t$ and the future frame $c_{t+h}$ as inputs to extract a latent code representing "dynamics" (removing static background information). Therefore, to reconstruct the future frame $c_{t+h}$, the decoder must explicitly receive the current frame $c_t$ as a supplement for the static information. This asymmetric design follows LAPA[2].
> 2. Motion Branch: The motion decoder is designed as a conditional module. We adopt the 'action expert' architecture from $\pi_0$[3], where the decoder is conditioned on proprioceptive states $m_t$. This design is strategic: it allows the motion decoder to be directly reused as the action policy head in Stages 2 and 3. Other approaches like Discrete Policy[4] condition on vision $o$ and language $l$ is also an option.
>
> [1] Zhu, Zixin, et al. "Designing a better asymmetric vqgan for stablediffusion." arXiv preprint arXiv:2306.04632 (2023).
>
> [2] Ye, Seonghyeon, et al. "Latent action pretraining from videos." Proceedings of the International Conference on Learning Representations (ICLR) (2025).
>
> [3] Black, Kevin, et al. "$\pi_0 $: A Vision-Language-Action Flow Model for General Robot Control." Proceedings of Robotics: Science and Systems (RSS) (2025).
>
> [4] Wu, Kun, et al. "Discrete policy: Learning disentangled action space for multi-task robotic manipulation." 2025 IEEE International Conference on Robotics and Automation (ICRA). IEEE, 2025.
>
> **[R2Q2]**
> > Justification on the vision-motion alignment.
>
> We appreciate the reviewer’s insightful question regarding the design rationale of our vision-motion alignment.
>
> 1. Our vision encoder takes both the current frame $c_t$ and the future frame $c_{t+h}$ as inputs to extract a latent code capturing visual dynamics, while the decoder explicitly utilizes the current frame $c_t$ to provide static environmental information (e.g., background) for reconstructing the future frame.
> And our goal is not to force the vision and motion codes to be identical, but rather to use the alignment loss as an auxiliary regularization to guide the vision encoder in distilling task-relevant dynamics.
>
> 2. Crucially, this alignment is designed to be unidirectional, optimizing the vision embedding to approach the motion embedding, but not vice versa. We treat the robot's motion data as a semantic "anchor" because it directly reflects the essential task dynamics, whereas visual inputs are often entangled with complex backgrounds and irrelevant static variations, as mentioned in UniVLA[1]. By anchoring the vision code to the motion code, we ensure that the vision encoder focuses on the critical dynamic information required for the task, effectively filtering out visual noise while maintaining the ability to reconstruct future frames.
>
> [1] Bu, Qingwen, et al. "Univla: Learning to act anywhere with task-centric latent actions." Proceedings of Robotics: Science and Systems (RSS) (2025).

---

### Official Review · Reviewer_H8iB · 2025-11-01

**Soundness:** 3
**Presentation:** 4
**Contribution:** 4
**Rating:** 4
**Confidence:** 4

**Summary:**

The paper proposes XR-1, a three-stage training approach for vision-language-action models (VLAs) that leverages heterogeneous data sources from both Internet-scale human videos and a mixture of different robot datasets to produce a new VLA that outperforms prior VLAs, such as $\pi_0$, RDT, UniVLA, and GR00T-N1.5. The authors also introduce Unified Vision-Motion Codes (UVMC), a novel dual-branch latent representation that jointly encodes visual information and motion information into discrete codes in a shared VQ-VAE cookbook. The three training stages consist of self-supervised learning of UVMC across both human videos from the Internet and diverse robot datasets, cross-embodiment UVMC pretraining that injects the learned representations into the policy's VLM backbone through learnable input tokens, and platform-specific post-training for downstream tasks. Extensive experimental evaluations across 6 robot embodiments, 120 tasks, and 120K rollouts demonstrate XR-1's superior performance over prior VLAs such as $\pi_0$, UniVLA, RDT, and GR00T-N1.5.

**Strengths:**

* The paper is well-written and easy to follow.
* The experiments assess policies over an extensive set of embodiments, tasks, and trials in the real world (6 robot embodiments, 120 tasks, 120K trials). The proposed XR-1 policy outperforms the next best prior method by a significant margin (e.g. +30% in absolute success rate over $\pi_0$ in both the Dual-Arm UR-5e and Tien Kung 2.0 task suites).
* The authors propose Unified Vision-Motion Codes (UVMC), a technically novel VQ-VAE-based latent representation that encodes both visual information and robot motion information in a joint latent space.
* XR-1's strong performance holds across various tasks and generalization settings, validating its efficacy and supporting the paper's claims.
* The paper presents extensive details on implementation, training data, additional experiments, and evaluations, with plenty of details in the appendix. The authors have clearly put significant effort into providing details that may be helpful for understanding the paper better and reproducing the method and experiments.

**Weaknesses:**

* A major weakness motivating my current recommendation for this paper is that comparing XR-1 to prior methods such as $\pi_0$, RDT, UniVLA, and GR00T-N1.5 in Sections 4.2 and 4.3 does not seem to be a fair comparison given that the former benefits from pretraining on a superset of tasks in the XR-D dataset that are similar to the embodiment-specific tasks that all the methods are post-trained and evaluated on (or provides support for the robot embodiments that are evaluated on after post-training, such as Dual-Arm UR-5e, which appears in both XR-D pretraining and in post-training). Perhaps a more fair direct comparison with the baseline methods would be a variant of XR-1 that is fine-tuned on task-specific datasets without XR-D pretraining. This would enable a more fair analysis of whether the XR-1 training recipe and UVMC representation are beneficial, since there is potentially a confounding factor in the performance improvement that comes from additional supervision from pretraining data collected on embodiments similar to the ones used for post-training evaluations. Currently, it is not clear whether the proposed training recipe or the XR-D dataset pretraining is contributing more to the performance. Further discussion on this or clarifications on the differences between train and test would be critical for adjusting my recommendation for this submission (I am willing to adjust my score since I believe there are several positive aspects to this paper otherwise).
* In the abstract, the authors write, "Existing methods often encode latent variables from either visual dynamics or robotic actions to guide policy learning, but they fail to fully exploit the complementary multi-modal knowledge present in large-scale, heterogeneous datasets." Similarly, in the related works, the authors write, "However, since the pre-training involves lagre-scale heterogeneous robotics and human data, existing VLA models remain limited ability to achieve effective unification across the heterogeneous modalities." These are fairly broad and abstract claims, and it is not clear what they mean and whether there is empirical validation for such claims. Clarification on the meaning behind these statements and why they are true (through supporting evidence) would strengthen the paper. For example, $\pi_0$ and OpenVLA have shown multi-task multi-embodiment control ability as well as adaptability to new robot embodiments through post-training, and it is not clear why XR-1 would be unique in these respects.
* In Section 4.2, the authors write, "Several baselines even collapse to 0% performance on harder tasks, which we attribute to insufficient auxiliary supervision and gradient conflicts during multi-task optimization." More details on this finding would strengthen the paper. Are prior methods struggling to fit the multi-task training dataset, or is there some other issue causing relatively poor performance compared to XR-1? Details on how baseline methods are trained and qualitative descriptions of how they fail would be helpful to aid understanding.
* There is no discussion or experiments on having two separate cookbooks for visual representations and motion representations. An ablation experiment showing that a unified cookbook is important would strengthen the paper.

**Questions:**

* In the "Lightweight Models" discussion in Section 4.4, in Table 4 rows (1) and (2), the difference between these two experiments is not clear. What is "FT." here and why does this correspond to "incorporating UVMC into Stage-3 training" as described in the main text? Perhaps some additional description in the table caption or main text can be helpful to the reader.

---

> ### Author Response · Authors · 2025-11-25
> **Response to Reviewer H8iB (Part 1)**
>
> We thank the reviewer for the insightful and constructive comments. We denote the weakness $n$ of Reviewer H8iB as R1Wn and question $n$ as R1Qn.
>
> **[R1W1]**
> > Concerns regarding unfair comparison due to pre-training data overlap.
>
> We thank the reviewer for the constructive feedback regarding the similar embodiment in pre-training and post-training. We address the concerns about fair comparison by providing the following evidence, clarification, and new experimental results:
>
> 1. Evidence of Embodiment Generalization (Tien Kung 2.0).
> To rigorously test generalization capability and rule out embodiment overlap, we have evaluated XR-1 on the Tien Kung 2.0 embodiment in Table 2, which is completely absent from the XR-D pretraining dataset.
> Thus, XR-1 is only fine-tuned using Tien Kung 2.0 data in stage 3.
> As shown in Table 2 of the manuscript, XR-1 achieves a strong success rate of 72% while the best baseline $\pi_0$ achieves a success rate of 40.8%. Furthermore, we have updated the table to include the performance of $\pi_{0.5}$ on this embodiment.
> Despite being a larger model, $\pi_{0.5}$ only achieves a 41.0% success rate. This significant margin unequivocally demonstrates XR-1’s superior capability for embodiment generalization, which stems from our proposed method rather than prior data exposure.
>
>
> 2. Performance Comparison on Dual-Arm UR-5e with Data Disparity.
> Regarding the Dual-Arm UR-5e configuration, we analyzed the pretraining distributions of the baselines. The $\pi_0$ pretraining dataset contains over 390 hours of Dual-Arm UR data (approx. 11M+ frames at 8Hz), which is more than double the amount found in our XR-D dataset (5.4M frames). The updated $\pi_{0.5}$ further expands this pretraining scale. To investigate this, we conducted comprehensive evaluations across 20 tasks on the Dual-Arm UR-5e setup in Figure 4 of the manuscript.
> As detailed in the newly added **Table R1_1** (below) and the updated manuscript with the results of $\pi_{0.5}$, XR-1 achieves an average success rate of 72%, substantially outperforming both $\pi_0$ (43%) and $\pi_{0.5}$ (62%). Even though the baselines had access to significantly more in-domain pretraining data for this specific configuration, XR-1 demonstrates superior performance. This proves the effectiveness of our proposed architecture and training strategy, rather than reliance on the scale of specific subsets within the pretraining data.
>
> Table R1_1: Success rate results across 20 tasks on Dual-Arm UR-5e.
> |     Method      | DUR-CloseToolBox   | DUR-CloseDoorKnob | DUR-TakeBasketTea | DUR-FindTapeBasket | DUR-StackBowls | DUR-HangCupHolder | DUR-PressButton | DUR-PickPlaceTape | DUR-OpenTrashBin | DUR-SweepRubbish |         |
> |--------------|--------------------|-------------------|-------------------|--------------------|----------------|-------------------|-----------------|-------------------|------------------|------------------|---------|
> | $\pi_0$      | 85                 | 85                | 80                | 70                 | 80             | 10                | 35              | 20                | 55               | 0                |         |
> | $\pi_{0.5}$  | 90                 | 50                | 90                | 45                 | 90             | 10                | 80              | 65                | 75               | 70               |         |
> | XR-1         | 90                 | 90                | 90                | 85                 | 85             | 85                | 85              | 85                | 85               | 80               |         |
> |              | DUR-FlipTennisTube | DUR-PickToolBox   | DUR-MoveCupMilk   | DUR-TransCupHolder | DUR-StackBrake | DUR-PlaceTools    | DUR-SweepTrash  | DUR-AssembleValve | DUR-TransButton  | DUR-StackCubes   | Average |
> | $\pi_0$      | 70                 | 70                | 55                | 15                 | 0              | 35                | 55              | 0                 | 20               | 15               | 43      |
> | $\pi_{0.5}$  | 70                 | 75                | 65                | 55                 | 70             | 65                | 60              | 45                | 35               | 20               | 62      |
> | XR-1         | 80                 | 75                | 65                | 65                 | 65             | 65                | 60              | 55                | 30               | 20               | 72      |

---

> ### Author Response · Authors · 2025-11-25
> **Response to Reviewer H8iB (Part 2)**
>
> **[R1W1]**
> > Concerns regarding unfair comparison due to pre-training data overlap.
>
> 3. Validity of "System vs. System" Comparison.
> We respectfully argue that a "System vs. System" comparison is the standard protocol for evaluating pretrained VLA models, rather than retraining every baseline on identical datasets. Retraining large-scale VLAs from scratch is computationally prohibitive, and pretraining datasets typically encompass the configurations found in downstream finetuning tasks. This methodology is consistent with recent state-of-the-art works.
> For instance, **RDT** is pretrained on a dataset where 9.9% of the data comes from Aloha Robot, and is subsequently finetuned and evaluated on Aloha (comparing against baselines without this specific pretraining). Similarly, **GR00T** includes massive amounts of GR-1 data in pretraining (6.4M teleop, 125.5M simulation, and 23.8M neural video frames) before being tested on the GR-1 robot against baselines that lack such extensive prior exposure. Therefore, our evaluation strategy aligns with the established community standards for foundational robotics models.
>
> **[R1W2]**
> > Unclear claims regarding the limitations of existing VLA models (e.g., OpenVLA) in exploiting heterogeneous data.
>
> We thank the reviewer for the constructive feedback. In the revised manuscript, we have revised Section 2 Related Work to explicitly clarify these distinctions and detail the specific limitations of prior arts (highlighted in red in the updated PDF).
> To address your concern regarding the uniqueness of XR-1 compared to models like OpenVLA, we summarize the key differentiators and the rationale behind our design below:
>
> 1. Most large-scale VLA methods (e.g., OpenVLA) follow a two-stage paradigm: pre-training on large-scale robotic data followed by fine-tuning on specific tasks. These models are typically tasked with directly generating precise, low-level 3D robotic actions from high-dimensional visual inputs. However, this end-to-end mapping is challenging to optimize as it lacks explicit guidance at the feature level. The model must implicitly learn to bridge the significant semantic gap between pixels and motor controls without intermediate guidance.
>
> 2. While some approaches introduce latent variables to mitigate the difficulty of direct mapping, they generally treat modalities in isolation:
>
> $\quad$ 2.1 Action-Centric: The first category focuses on modeling low-level motor dynamics by discretizing robotic actions into a sequence of discrete latent tokens.
>
> $\quad$ 2.2 Vision-Centric: The second category seeks to learn representations solely by exploiting the abundance of unlabeled video data.
> Crucially, by decoupling vision and action, these unimodal paradigms fail to fully capture the internal link between observation (what is seen) and execution (what is done).
>
> 3. XR-1 distinguishes itself by addressing the above limitations through a unified, multi-stage framework, especially in stage 1:
>
> $\quad$ 3.1 Latent Feature Learning: Unlike standard VLAs, XR-1 introduces a self-supervised UVMC (Unified Vision-Motion Codes) learning stage. By synergizing human video data with large-scale robotic data, the model learns a highly expressive latent space that captures rich semantic information before action generation begins.
>
> $\quad$ 3.2 Cross-Modal Alignment: XR-1 extracts discrete latent variables for both motion and vision. We introduce a cross-modal alignment loss that explicitly enforces consistency between visual dynamics and robotic motion. This alignment ensures that the learned features are not just representationally rich but also causally aligned, significantly enhancing the model's ability to generate precise actions in stage 3.

---

> ### Author Response · Authors · 2025-11-25
> **Response to Reviewer H8iB (Part 3)**
>
> **[R1W3]**
> > More analysis/details on baseline training and failures.
>
> We appreciate the constructive feedback regarding baseline comparisons.
> We provided detailed hyperparameter settings for baselines as follows, and also provided a comprehensive failure analysis in Appendix 6.8 of the updated PDF (highlighted in red).
> A summary of these implementation details and our findings follows.
>
> 1. Implementation Details of Baselines.
> To ensure a rigorous comparison, we trained all baselines for approximately 40 epochs to ensure convergence.
>
> $\quad$ 1.1 UniVLA: We employed LoRA fine-tuning as full fine-tuning resulted in Out-Of-Memory (OOM) errors. Config: Batch size 8 A100 GPUs $\times$ 6; Chunk size 16.
>
> $\quad$ 1.2 GR00T-N1.5: Full fine-tuning was performed. Config: Batch size 8 A100 GPUs $\times$ 12; Chunk size 16.
>
> $\quad$ 1.3 $\pi_0$: We froze the vision encoder and the language encoder. Config: Batch size 8 A100 GPUs $\times$ 4; Chunk size 50.
>
> $\quad$ 1.4 RDT: We froze the vision encoder and the language encoder. We also added action normalization to stabilize training. Config: Batch size 8 A100 GPUs  $\times$ 32; Learning rate 2e-5; Chunk size 64; Image history size $2$.
>
> $\quad$ 1.5 XR-1 (Ours): In stage-3, we froze the vision encoder and the language encoder. Config: Batch size 8 A100 GPUs $\times$ 20; Learning rate 5e-5; Chunk size 50.
>
> 2. Analysis of Failure Cases.
> We conducted a qualitative review of the rollout videos to investigate why baselines struggle compared to XR-1 and added the analysis to Appendix 6.8 (highlighted in red in the updated pdf). We categorize the observed failures into two primary modes:
>
> $\quad$ 2.1 Optimization Collapse & Conflicting Gradients
> In some distinct scenarios, baselines fail to capture the correct motion trend entirely. The models exhibit "hesitation" or revert to the mean pose, suggesting that the optimization objective is torn between conflicting gradients from different tasks.
> Example: In the `DFR-CloseToolbox` task, the $\pi_0$ policy initiates a downward movement with the right arm but immediately retracts to the initial position. The robot appears indecisive and fails to commit to the task trajectory. We attribute this to the difficulty of fitting a single policy distribution to 20 diverse tasks without task-distinguishing representations.
>
> $\quad$ 2.2 Precision Deficiency & Coordination Failure
> Another common failure mode involves the robot attempting the correct action but failing in execution precision or bimanual coordination.
> Example: In the `DUR-TransCupHolder` task, GR00T-N1.5 successfully grasps the cup with the right arm. However, it drops the cup during the handover to the left arm. This indicates that while the model learns the general policy distribution, it lacks the fine-grained control and temporal consistency required for complex, multi-stage manipulation.
>
> $\quad$ 2.3 XR-1 addresses these issues through the Unified Vision-Motion Condition (UVMC) introduced in Stage 1.
> The UVMC serves as a compact representation of visual dynamics and motion patterns. By conditioning the policy on UVMC, XR-1 can explicitly distinguish between different task modes, thereby reducing gradient conflicts during multi-task optimization.
> As an intermediate feature supervision signal, UVMC guides the model to generate smoother and more physically consistent actions. This additional supervision is critical for tasks requiring high precision (e.g., dual-arm handover), preventing the coordination failures observed in baselines like GR00T.

---

> ### Author Response · Authors · 2025-11-25
> **Response to Reviewer H8iB (Part 4)**
>
> **[R1W4]**
> > Ablation studies on separate visual and motion cookbooks.
>
> We appreciate the constructive suggestion to investigate the impact of using a separate codebook. To address this, we conducted an ablation study comparing our default shared codebook design against a separate codebook variant. Since modifying the codebook architecture precludes the reuse of existing pre-trained checkpoints, we trained the model (Stages 1–3) from scratch using downstream task data.
> As presented in **Table R1_2**, we compared the success rates of both configurations across 6 tasks on the Dual-Arm UR-5e embodiment. The results indicate that both approaches yield comparable performance. We attribute this similarity to the fact that while a separate codebook increases the total number of learnable codes, the inclusion of an alignment loss effectively constrains the relationship between modalities, resulting in similar execution capabilities.
> Furthermore, in response to Reviewer vfX7's first weakness comment, we have conducted additional ablation studies to evaluate performance when utilizing solely the vision code or the motion code during inference.
> We will integrate these comprehensive ablation studies into the appendix of the revision.
>
> Table R1_2: Success rate results across 6 tasks on Dual-Arm UR-5e. "DT" indicates training directly on the downstream task data.
> | sepe codebook | latent token | stage-1 | stage2 | stage3 | DUR-CleanTable | DUR-FindTapeBasket | DUR-MoveCupMilk | DUR-StackBowls | DUR-SweepTrash | DUR-TransCupHolder | Avg.  |   |
> |---------------|--------------|---------|--------|--------|----------------|--------------------|-----------------|----------------|----------------|--------------------|-------|---|
> | T             | both         | DT      | DT     | DT     | 55             | 75                 | 55              | 85             | 40             | 80                 | 65.0    |   |
> | F             | motion       | DT      | DT     | DT     | 20             | 35                 | 35              | 55             | 15             | 25                 | 30.8  |   |
> | F             | vision       | DT      | DT     | DT     | 30             | 40                 | 45              | 70             | 55             | 60                 | 50.0    |   |
> | F             | both         | DT      | DT     | DT     | 50             | 75                 | 65              | 80             | 60             | 70                 | 66.7  |   |
>
> **[R1Q1]**
> > Additional description for Table 4.
>
> We appreciate the constructive feedback and have revised the caption and main text associated with Table 4 (highlighted in red) to clarify these experimental settings. In Table 4, "DT" denotes training with downstream task data.
> Compared to row (1) in Table 4, row (2) involves training Stage-1 and Stage-2 on the downstream task data.

---

### Author Response · Authors · 2025-12-03
**Global Response [update of PDF & Point-by-Point Reply] (Part 1)**

First and foremost, we would like to express our sincere gratitude to the Reviewers, Area Chairs, and Program Chairs for their time and dedicated effort. We have carefully studied all comments and have provided detailed **point-by-point** responses under each Reviewer's section.

We are encouraged not only by the positive scores but, more importantly, by the recognition of our work from the reviewers across four key dimensions:

**Method**: "a technically novel VQ-VAE-based latent representation" `Reviewer H8iB`; "The proposed method is novel" `Reviewer PrUm`; "The technical quality and empirical rigor are exceptional." `Reviewer vfX7`; "The idea is both novel and interesting." `Reviewer gGq7`.

**Evaluation**: "The experiments assess policies over an extensive set of embodiments, tasks, and trials in the real world (6 robot embodiments, 120 tasks, 120K trials)." `Reviewer H8iB`; "The paper presents extensive real-world experiments, validating XR-1 with over 12,000 rollouts across six different robot embodiments." `Reviewer PrUm`; "The paper is grounded by an immense evaluation campaign." `Reviewer vfX7`; "Comprehensive experiments effectively demonstrate the superior performance of the proposed method" `Reviewer gGq7`.

**Presentation**: "The paper is well-written and easy to follow." `Reviewer H8iB`; "This paper is well-presented and easy to follow. The motivations are clear and insightful, and the methodology is straightforward to understand." `Reviewer gGq7`.

**Performance**: "XR-1 shows strong performance across diverse tasks and generalization settings." `Reviewer H8iB`; "demonstrating strong cross-task and cross-embodiment generalization." `Reviewer PrUm`; "XR-1 consistently outperforms strong recent baselines (e.g., GR00T-N1.5) across bimanual, dexterous, and long-horizon scenarios." `Reviewer vfX7`; "Comprehensive experiments clearly demonstrate the superior performance of the proposed method." `Reviewer gGq7`.

In response to the questions, we have added more **rigorous and comprehensive analyses**, including:
- Expanded UVMC ablation studies
- Evaluations of the impact of human video pre-training
- Visualization-based interpretability studies
- Detailed baseline and XR-1 failure case analyses

These updates, along with refined terminology, are detailed in the revised PDF (highlighted in red). ***A summary of these changes*** follows:

Section 2.1/2.2 (**Related work**): To address `Reviewer H8iB`'s concern about **unclear claims regarding the limitations of existing VLA models**, we have revised Section 2 to explicitly clarify our distinctions and more clearly articulate the specific limitations of prior work.

Section 4.2 (**Main Results**): To address `Reviewer H8iB`'s concern about potentially **unfair comparisons due to pre-training data overlap**, we have **added an additional baseline, $\pi_{0.5}$,** and evaluated it across six different robot embodiments. We further show that XR-1 **achieves SOTA performance on the Tienkung 2.0** embodiment, which is **not included in the XR-D pretraining** set, and that despite **$\pi_0/\pi_{0.5}$ being pretrained with substantially more Dual-Arm UR-5e data, XR-1 still performs better**, indicating that our advantage stems from the proposed architecture and training strategy rather than from pre-training data overlap.

Appendix 6.6 (**Additional Ablation study**): To address `Reviewer H8iB, vfX7, and gGq7`'s request for **more ablation studies of UVMC**, we have added Table 14, which **includes 10 groups of experiments analyzing the UVMC architecture and hyperparameters**. These experiments examine (i) separate versus combined visual and motion codebooks, (ii) alternative representation learning paradigms that use only visual-dynamics latents or only action latents, and (iii) the impact of different codebook hyperparameter settings.
To address Reviewer vfX7 and gGq7’s questions regarding the impact of human video data (Ego4D), we have added Table 15, which evaluates how mixing human video data with robot data during the UVMC pre-training stage affects the final performance.

Appendix 6.7 (**Visualization**): To address `Reviewer PrUm and vfX7`'s requests for **more interpretability and a deeper analysis of the unified vision-motion space**, we analyze **nearest-neighbor retrieval in Section 6.7.1 and add t-SNE visualizations in Section 6.7.2**. These visual analyses focus on the semantic similarity between vision and motion modalities and illustrate that the learned representations are largely embodiment-agnostic.

Appendix 6.8/6.9 (**Failure Case Analysis**): To address `Reviewer H8iB and vfX7`'s request for **more analysis of baseline and XR-1 failure cases**, we provide a **comprehensive baseline failure analysis** in Appendix 6.8 and investigate XR-1’s failure cases in detail in Appendix 6.9.

---

### Author Response · Authors · 2025-12-03
**Global Response [update of PDF & Point-by-Point Reply] (Part 2)**

Appendix 6.10 (**Simulation Benchmark**): To address `Reviewer PrUm`’s suggestion that evaluation on **a standard public benchmark would provide a more comprehensive assessment**, we have added simulation experiments on a simpler platform, which show a **27% performance gain** compared to $\pi_0$.

Minor Corrections & Terminology:
We corrected spelling errors in Figure 2 and citation errors in Table 3, and refined the use of the term “cognitive-inspired” throughout the paper to avoid ambiguity.

In particular, regarding `Reviewer PrUm`’s **concern about comprehensive assessment and reproducibility**, we respectfully note that we conducted more than **14,000 real-world evaluations**, substantially exceeding the scale of recent leading works such as $\pi_0$ (20 tasks with 660 rollouts), RDT (7 tasks with 875 rollouts), and GR00T (13 tasks with 520 rollouts). We **will also release our code and checkpoints**. In addition, following the reviewer’s suggestion, we **have added simulation experiments on a simpler platform, which show a 27% performance gain**. We believe this methodology provides distinct value to the ML community by demonstrating how latent-space learning, combined with a three-stage training paradigm, can outperform standard two-stage training in multi-task, multi-embodiment, and diverse environments.

We sincerely hope to engage in constructive dialogue with reviewers as we believe this exchange is vital for improving the quality of our work.

Best regards,
Authors of Submission 15019

---

### Meta-Review · Area_Chair_zEfw · 2026-01-07

**Summary:**

The major concerns raised several reviewers are about the fairness of comparison with existing baselines.

In rebuttal, the authors provide some simple simulation experiments and more comparison with pi0/pi0.5.

My final recommendation is rejection since indeed though with extra experiments, I still do not agree that the comparison is totally fair for baselines. For simulation results, the experiments are over simplified. Since there are no clues regarding open source, such unclear comparison (with lots of unknown factors) makes it difficult for community to reproduce and significantly limits its impact.

**Reviewer Concerns:**

The concerns regarding the implementation and writings are solved.

The concerns regarding the fairness compared to existing baselines are not fully solved.

**Reviewer Scores:**

Reviewer H8iB and PrUm might keep their negative scores as the fair comparison with baselines are not fully solved.

Reviewer vfX7 and gGq7 might keep their positive scroes.

---

### Decision · Program_Chairs · 2026-01-26

Reject